# TamperBench: Systematically Stress-Testing LLM Safety Under Fine-Tuning and Tampering

## Abstract

As increasingly capable open-weight large language models (LLMs) are deployed, improving their *tamper resistance* against unsafe modifications, whether accidental or intentional, becomes critical to minimize risks. However, there is no standard approach to evaluate tamper resistance. Varied data sets, metrics, and inconsistent threat settings make it difficult to compare safety, utility, and robustness across different models and defenses. To this end, we introduce TamperBench, the first unified framework to systematically evaluate the tamper resistance of LLMs. TamperBench (i) curates a repository of weight-space fine-tuning attacks and latent-space representation attacks; (ii) allows for testing state-of-the-art tamper-resistance defenses; and (iii) provides both safety and utility evaluations. TamperBench requires minimal additional code to specify any fine-tuning configuration, alignment-stage defense method, and metric suite while ensuring end-to-end reproducibility. In this work, we use TamperBench to evaluate 21 open-weight LLMs, including defense-augmented variants, across nine tampering threats using standardized safety and capability metrics with hyperparameter sweeps per model-attack pair. *Code is available at*: https://anonymous.4open.science/r/TamperBench-71DD

## 1 Introduction

Diverse training procedures are used to safety-align modern LLMs (Touvron et al., 2023; OpenAI et al., 2024; Team et al., 2023), yet *tampering*, modifications to the model's weights or latent representations, can undermine these safeguards (Che et al., 2025; Huang et al., 2024b; Qi et al., 2024b; Murphy et al., 2025; Halawi et al., 2024; Schwinn & Geisler, 2024). This threat is increasingly pressing as compute-efficient approaches such as LoRA (Hu et al., 2022; Zhao et al., 2024; Meng et al., 2024) make tampering accessible at low cost, and open-weight models—modifiable by anyone—have capabilities closely behind those of frontier models (Cottier et al., 2024), which are already classified as high-risk under frontier safety frameworks (OpenAI, 2025; Anthropic, 2025).

In response to such tampering risks, over twenty defenses have been proposed in the past year alone (Lyu et al., 2024; Hsu et al., 2024; Tamirisa et al., 2025; Shen et al., 2025; Li et al., 2025b;a; Huang et al., 2025; Zhao et al., 2025; Bianchi et al., 2024; Qi et al., 2025; Casper et al., 2024; Wang et al., 2024a; Huang et al., 2024c;d; Liu et al., 2025; Huang et al., 2024a; Liu et al., 2024; Mukhoti et al., 2024; Wei et al., 2024; Du et al., 2025; Sheshadri et al., 2025; O'Brien et al., 2025; Simko et al., 2025) reflecting an active and fast-moving research landscape.

Despite the proliferation of proposed defenses, evaluations are fragmented as works differ in their choice of attacks, threat models, and metrics (Figure 2). The field lacks a standard basis for determining to what degree a defense improves tamper resistance (Huang et al., 2024b; Qi et al., 2024a). Casper et al. (2025) identify "model tampering evaluations" as a central open problem, noting that current work lags in testing models against a diverse suite of tampering attacks and hyperparameter settings. A comprehensive, standardized evaluation is necessary for understanding which tampering defenses are most promising, whether tamper resistance is tractable at all, and what precautions developers should take in releasing highly capable open-weight models.

To address this gap, we introduce TamperBench (Figure 1), the first systematic benchmark and toolkit for evaluating tamper resistance in open-weight LLMs. TamperBench provides an extensible suite of tampering attacks and standardized evaluation protocols, together with interfaces that make it

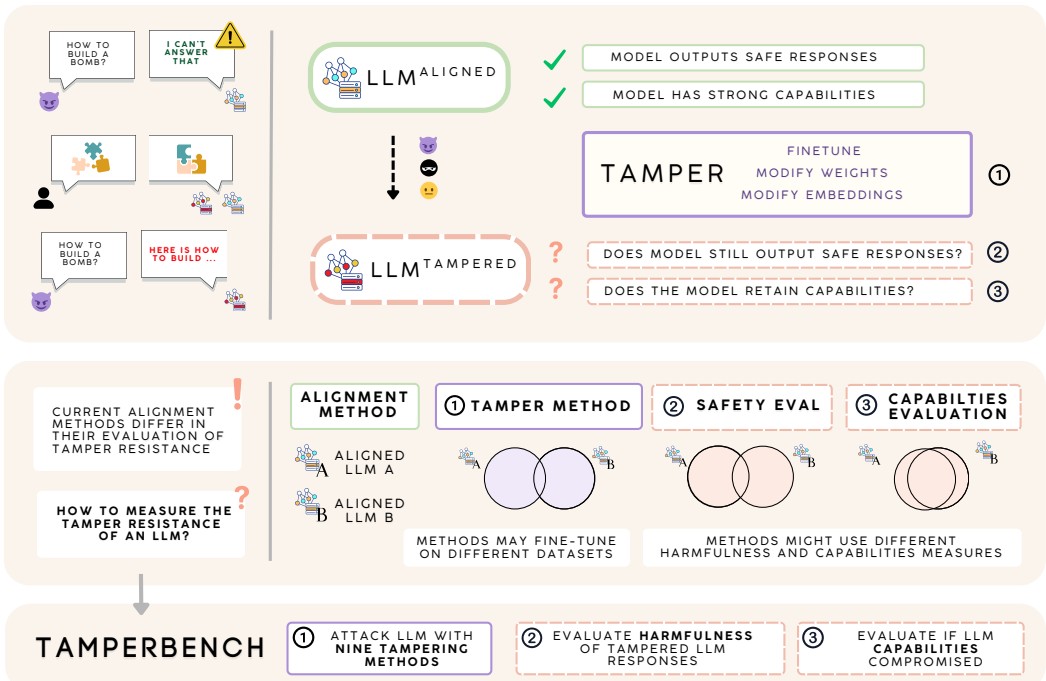

Figure 1: Tampering LLMs, as defined by Che et al. (2025)), involves modifying their weights or latent representations and can compromise safety guardrails, yielding models that can output harmful responses. While numerous methods have been proposed to make models 'tamper-resistant', there is a lack of a systematic framework to measure this. TAMPERBENCH provides a framework to stress test LLM safety under tampering.

straightforward to integrate and test defenses. The framework covers both benign and adversarial tampering regimes, including overt fine-tuning attacks and stealthy manipulations designed to evade closed-weight moderation systems. It supports both weight-space modifications and latent-space perturbations at inference time, enabling a unified view of diverse tampering approaches.

The framework integrates with modern toolkits including vLLM, Transformers, and Optuna, enabling efficient large-scale experimentation, systematic hyper-parameter sweeps, and multi-GPU execution. In addition to standardized safety metrics (StrongREJECT; Souly et al., 2024) and capability benchmarks (MMLU-Pro; Hendrycks et al., 2021), TAMPERBENCH allows users to analyze both harmfulness and utility after tampering, offering a more complete picture of model behavior beyond binary safeguard bypass.

**Our contributions can be summarized as follows:**

- **Open-Source Benchmark and Toolkit:** We introduce TAMPERBENCH, a unified open-source benchmark and toolkit for evaluating tamper resistance in open-weight LLMs. The field currently lacks a standardized basis for determining whether robustness is actually improving. TAMPERBENCH fills this gap by consolidating tampering attacks, evaluation protocols, and defense interfaces into a single extensible framework, enabling reliable and comparable assessments of open-weight LLMs and tamper-resistance defenses.

- **Realistic Adversarial Evaluation:** We perform hyperparameter sweeps for attack–model pairs to reflect realistic adversarial conditions, reducing sensitivity to arbitrary training choices and enabling robust comparisons of susceptibility across attacks and models.

- **Comparative Analysis of Open Models:** Using TAMPERBENCH, we evaluate 21 open-weight LLMs—including base, instruction-tuned, and defense-augmented variants—across nine tampering attacks with standardized safety and capability metrics. This reveals family-level patterns: (i) post-trained Llama-3 models are more vulnerable than their base counterparts; (ii) Qwen3 base models are more tamperable than their post-trained variants; and (iii)

| ⚠ SAFETY EVALUATION | | | | |
| --- | --- | --- | --- | --- |
| **DEFENSE** | **HARMBENCH** | **BEAVERTAILS** | **STRONGREJECT** | **GPT-4 JUDGE** |
| **TAR** TAMIRISA ET AL., 2025 | ✓ | ✗ | ✗ | ✗ |
| **VACCINE** HUANG ET AL., 2024D | ✗ | ✓ | ✗ | ✗ |
| **RR** ZOU ET AL., 2024 | ✓ | ✗ | ✗ | ✗ |
| **LAT** SHESHADRI ET AL., 2025 | ✓ | ✗ | ✓ | ✓ |

| 🎨 BENIGN CAPABILITIES EVALUATION | | | | |
| --- | --- | --- | --- | --- |
| **ALIGNMENT STAGE DEFENSE** | **MT-BENCH** | **MMLU** | **OPENLLM** | **SST2** |
| **TAR** TAMIRISA ET AL., 2025 | ✓ | ✗ | ✗ | ✗ |
| **VACCINE** HUANG ET AL., 2024D | ✗ | ✓ | ✗ | ✓ |
| **RR** ZOU ET AL., 2024 | ✓ | ✗ | ✓ | ✗ |
| **LAT** SHESHADRI ET AL., 2025 | ✓ | ✗ | ✗ | ✗ |

Figure 2: While many alignment stage defenses have been proposed Tamirisa et al. (2025); Huang et al. (2024d); Zou et al. (2024); Sheshadri et al. (2025), they do not share any standardized evaluation. The absence of a standardized protocol means that defenses have not been fairly compared. This motivates TamperBench as the first framework to consolidate tampering attacks and evaluations into a unified toolkit.

within the 7–8B range, Qwen3-8B is marginally more resilient than Llama-3.1-8B-Instruct, whereas Mistral-7B-Instruct exhibits comparatively higher susceptibility to tampering.

## 2 BACKGROUND AND RELATED WORKS

### 2.1 LLM VULNERABILITIES

Large language models (LLMs) are typically aligned through supervised fine-tuning (SFT) (Wei et al., 2022) and reinforcement learning from human feedback (RLHF) (Ouyang et al., 2022; Casper et al., 2023), but are often adapted further in ways that compromise the alignment. Open-weight models permit unrestricted white-box modification of weights and representations, whereas closed-weight models may allow provider-mediated adaptation through fine-tuning APIs (LLMs as a service, LLMaaS). Parameter-efficient methods such as LoRA (Hu et al., 2022) and related adapters (Rajabi et al., 2025; Zhao et al., 2024; Meng et al., 2024) further reduce the cost of such modifications. Yet safety is typically evaluated only on the original aligned model, underestimating the harmfulness possible after tampering(OpenAI, 2024; Meta, 2025).

Fine-tuning has shown to be able suppress refusals with only a few harmful examples (Qi et al., 2024b; Che et al., 2025; Poppi et al., 2025), and even benign fine-tuning can destabilize safeguards (He et al., 2024; Pandey et al., 2025; Hu et al., 2025). Additionally, models can be tampered with in a more covert manner to bypass existing moderation safeguards in closed weight models, by embedding hidden behaviors through backdoors, mixing in a small proportion of harmful data with benign data, and re-constructing harmful data (Davies et al., 2025; Halawi et al., 2024; Murphy et al., 2025). Other work operates directly in representation space, by adapting latent space embeddings to elicit harmful responses or ablating refusal directions (Arditi et al., 2025; Schwinn & Geisler, 2024). TAMPERBENCH implements these tampering attacks so that it can measure the post-tampering harmfulness of open-weight models.

### 2.2 TAMPERING DEFENSES

To address vulnerabilities induced by tampering attacks, defenses aim to: (i) minimize *harmfulness* of model responses after adversarial attacks and (ii) maintain *utility* on benign tasks. Harmful-response

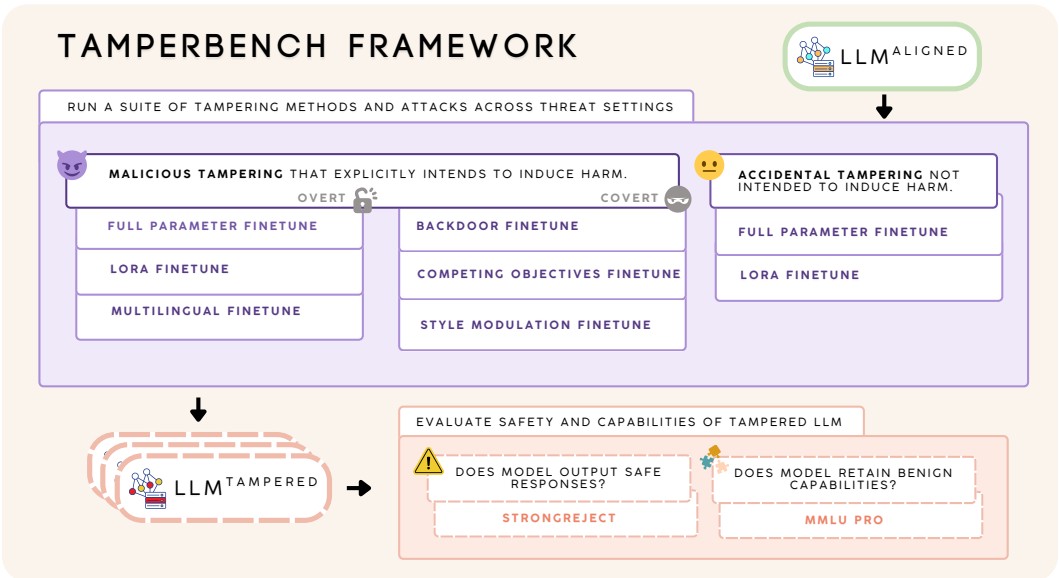

Figure 3: TamperBench evaluates tampering based on an actor or user's intent: malicious or benign (accidental). In addition, attacks could be covert if they are designed to bypass closed-weight moderation safeguards. These are represented by attacks such as harmful and benign fine-tuning, LoRA, multilingual tuning, backdoors, style modulation, competing-objectives, and embedding attacks. Models are assessed on safety with StrongREJECT and on benign capabilities with MMLU-Pro.

rates are often scored with LLM judges (Wang et al., 2024a; Qi et al., 2024b), while stability is measured by task accuracy on standard benchmarks (Huang et al., 2024d; Li et al., 2025a).

Defenses are grouped according to the stage of intervention in the training pipeline. (1) *Alignment-stage defenses* strengthen the base model before it is made available to third parties by modifying the safety training process, such as by incorporating adversarial objectives, unlearning behaviors or simulating fine-tuning steps (Golatkar et al., 2020a;b; Henderson et al., 2023; Tamirisa et al., 2025; Zhao et al., 2025; O'Brien et al., 2025). Defenses at this stage are not mutually exclusive with other stages, and are thus the most appliable. (2) *Fine-tuning-stage defenses* modify adaptation dynamics through curated alignment data or auxiliary losses (Huang et al., 2024c; Wang et al., 2024a; Du et al., 2025; Sheshadri et al., 2025). (3) *Post-tuning defenses* repair misalignment after tampering via adversarial realignment or surgical weight edits (Hsu et al., 2024; Huang et al., 2024a).

Defense categories (2) and (3) above presuppose centralized control over fine-tuning, making them primarily applicable for commercial LLMaaS providers. By contrast, open-weight models are widely redistributed and adapted without oversight, leaving no mechanism for providers to enforce fine-tuning or post-tuning defenses. This makes tamper resistance for open weights a particularly pressing open challenge. Alignment-stage defenses (category 1) are the only strategies that embed durability directly into the base model, and thus remain relevant across both open-weight and API-based deployments. For this reason, our benchmark emphasizes systematic evaluation of alignment-stage defenses, while still supporting integration of categories (2) and (3) for completeness.

## 2.3 EXISTING FRAMEWORKS

Popular frameworks such as HarmBench (Mazeika et al., 2024) focus on automated red-teaming and refusal robustness. Yet they are confined to prompt-based attacks (jailbreaks, persuasion, harmful queries) and do not systematically evaluate weight-space tampering or fine-tuning regimes. These overlooked regimes pose equally critical threats, as they directly modify model parameters and can erode refusal behaviors in ways jailbreak-style prompting cannot capture. Current toolkits focused on benchmarking tamper resistance (Wang et al., 2024a; Qi et al., 2024b; Murphy et al., 2025) remain limited in extensibility, ease of onboarding new defenses, coverage of tampering regimes, and integration of diverse strategies. Huang et al. (2024b) argue "It is imperative to create a standard

benchmark". The gap in the resources and understanding for rigorous evaluation of tamper-resistance has led to unstable and overturned conclusions (Qi et al., 2024a). TAMPERBENCH fills this gap by unifying tampering attacks, defenses, and evaluation metrics, enabling reproducible and comparable assessment of resistance and stability across both weight- and latent-space manipulations.

# 3 TAMPERBENCH FRAMEWORK

## 3.1 THREAT MODEL

To reason about LLM threats systematically, we consider an actor's (1) *intent* and (2) *access*. An actor may tamper with (e.g., fine-tune) a model for benign goals or with explicitly malicious aims of weakening safeguards. They may have access to open-weight checkpoints or to provider fine-tuning APIs, and while TAMPERBENCH primarily targets open-weight threats, many attacks are designed to evade API-level moderation and thus pose risks in both settings. In our threat model, a model is *successfully tampered* if its safeguards are weakened (harmful responses increase) while its general capabilities are largely preserved, since models that lose utility are less attractive to adversaries. Defenders, in turn, seek to limit increases in harmfulness while preserving benign capabilities.

*Accidental misuse* arises when developers fine-tune an aligned model for ostensibly benign adaptation but inadvertently erode refusal behaviour and cause harmful responses to re-emerge (Qi et al., 2024b; Che et al., 2025; He et al., 2024). Here the (1) *intent* is to improve performance on a target application, and (2) the actor uses standard fine-tuning *access* (data and hyperparameter choices) in both open- and closed-weight settings; the resulting *risk* is that safety degrades as an unintended side effect.

*Malicious tampering* covers both overt and covert attempts to weaken safeguards. In both cases, the actor's (1) *intent* is to induce harmful or unrestricted behaviour, but (2) their *access* shapes how the attack is designed. Overt attacks assume unrestricted white-box access and therefore directly modify model weights or representations, such as through harmful or multilingual fine-tuning. Covert attacks, by contrast, are designed to operate under more restrictive access (e.g., fine-tuning APIs) and embed harmful behaviours in ways intended to bypass moderation or detection. In TAMPERBENCH, both forms are evaluated in the open-weight setting for comparability: overt attacks use unconstrained fine-tuning, whereas covert attacks are implemented using setups designed to maintain stealthiness.

## 3.2 TAMPER ATTACK SUITE

Within this threat-model framework, TAMPERBENCH instantiates tampering via a suite of weight-space and representation-space attacks (Figure 3). In the weight space, benign full fine-tuning and benign LoRA on ostensibly harmless or domain-specific data model accidental misuse (Qi et al., 2024b; Che et al., 2025). Harmful full fine-tuning, harmful LoRA, and multilingual fine-tuning (Poppi et al., 2025) on jailbreak or uncensored datasets capture overt malicious tampering (Che et al., 2025). Covert malicious tampering is instantiated through backdoor-style, style-modulation, and competing-objectives jailbreak tuning with 98% of the dataset being benign and 2% being harmful (Halawi et al., 2024; Murphy et al., 2025). In the representation space, latent embedding attacks perturb internal representations or inject poisoning signals while preserving benign behaviour but enabling harmful completions under hidden triggers (Schwinn & Geisler, 2024), providing a complementary axis of stealthy tampering.

## 3.3 UTILITY EVALUATION

TAMPERBENCH evaluates model utility on the MMLU-Pro (Wang et al., 2024b) dataset, measuring accuracy across 14 subject areas. Compared to the original MMLU (Hendrycks et al., 2021) dataset, MMLU-Pro introduces more challenging, reasoning-focused questions with an expanded choice set from four to ten options and improves the dataset quality. We evaluate benign capabilities on a MMLU-Pro validation set using a 5-shot chain-of-thought (CoT) prompt. This setup enables assessment of whether tampering attacks or defenses impair a model's core capabilities.

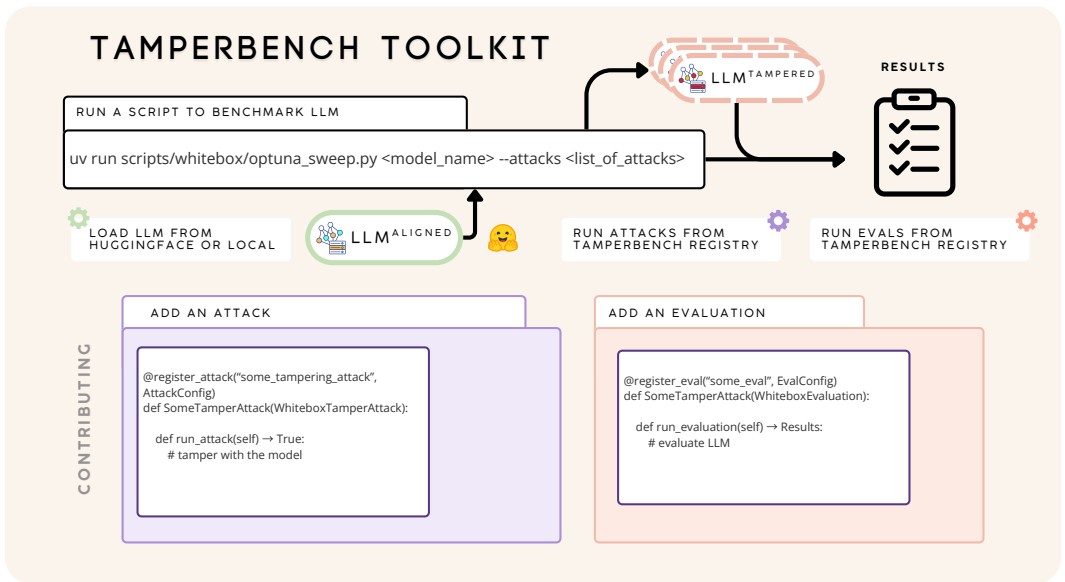

Figure 4: A single script can be run to benchmark an LLM by providing either a local checkpoint path or a HuggingFace repository ID, along with a list of attack names. The toolkit then executes the specified tampering attacks and evaluation modules, producing results scored with standardized safety and utility metrics and cached for reproducibility. TamperBench is designed to be highly extensible; enabling researchers to contribute methods with minimal code overhead.

## 3.4 SAFETY EVALUATION

To quantify residual harmful behavior, we employ JailbreakBench (Chao et al., 2024) and the StrongREJECT (Souly et al., 2024) dataset and evaluator. For embedding-based tampering attacks, we use JailbreakBench, which provides ground-truth model outputs under adversarial prompting. This allows us to directly evaluate whether tampering increases a model's propensity to produce unsafe completions. To assess the safety of responses more generally, we use the StrongREJECT evaluator (Souly et al., 2024), which achieves state-of-the-art agreement with human annotations, outperforming alternative classifier-based safety evaluators. For each prompt-response pair, it assigns each prompt-response pair a score between 0.0 and 1.0, where higher scores indicate more harmfulness.

## 3.5 TAMPERBENCH TOOLKIT

TAMPERBENCH's core registry provides unified interfaces for ALIGNMENT DEFENSES, ATTACKS, and EVALUATIONS. Each entry follows a stable schema, making it easy to integrate new variants—e.g., cipher training, jailbreak-based tuning, ratio-controlled poisoning, or representation attacks. Benchmarks run directly on HuggingFace models with multi-GPU support if needed. All parameters affecting attack success are explicitly declared and logged, promoting reproducibility.

Modular helpers support both end-to-end pipelines (*attack → train → evaluate*) and independent use of attacks or evaluations. Built-in `Optuna` integration enables efficient hyperparameter search for defenses and training regimes, while standardized logging and checkpointing ensure robust experimentation. Further, this design supports systematic sweeps over attack scenarios and evaluations, enabling controlled comparisons without ad-hoc scripts.

## 4 BENCHMARK EXPERIMENTS

We evaluate **21** open-weight LLMs spanning compact and mid-scale regimes (0.6B to 8B parameters), including both base and instruction-tuned variants.

The suite spans strongly safety-aligned models (notably the Llama family) as well as models with weaker or less certain alignment training (such as the Mistral and Qwen families). We also evaluate

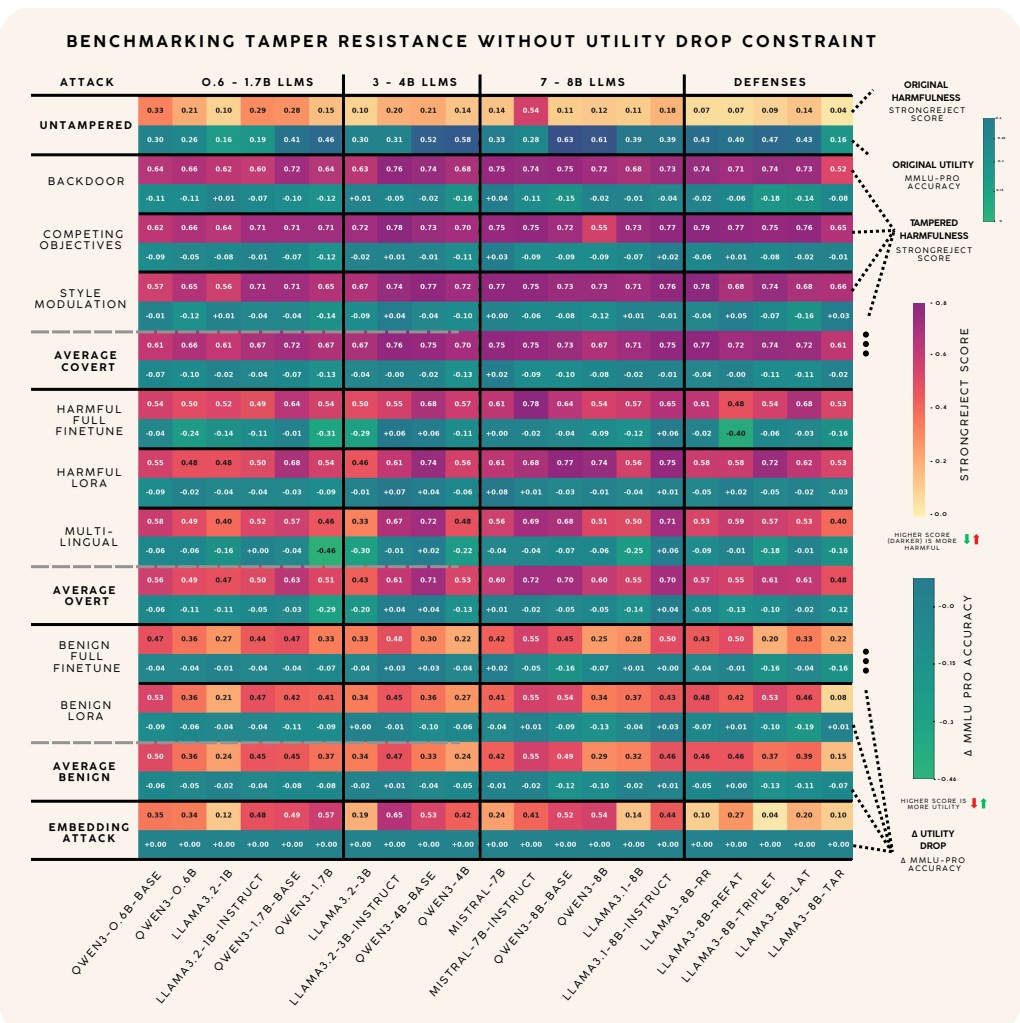

Figure 5: Benchmarking the tamper resistance of 21 large language models (LLMs) across a suite of attacks. Each cell reports a model's harmfulness, followed by its change in benign capabilities (MMLU-Pro). For each model–attack pair, hyperparameter sweeps are run and the highest (harmfulness) StrongREJECT is reported. Every model is vulnerable to a significant decrease in safety under all tampering settings.

five *defense-augmented* variants of Llama-3-8B-Instruct: (i) ReFAT (Yu et al., 2025), which simulates refusal-ablation tampering during training; (ii) Circuit Breaking (Zou et al., 2024; 2025), which disrupts harmful internal circuits; (iii) Triplet (Simko et al., 2025), which extends circuit breaking with contrastive representation learning; (iv) TAR (Tamirisa et al., 2025) which uses adversarial training & meta-learning techniques to build safeguards and (v) LAT (Casper et al., 2024) which leverages adversarial latent perturbation attacks in training.

For each *fine-tuning* attack, we run an Optuna-based hyperparameter search with 40 trials. We report the trial that maximizes StrongREJECT score, i.e., that provides maximum harmful assistance to an attacker (Appendix §A.7). Figure 5 presents, for each model–attack pair, harmfulness (StrongREJECT score) and change in benign capability (MMLU-Pro accuracy).

## 4.1 GLOBAL EFFECTS OF TAMPERING

At a high level, Figure 5 shows a consistent pattern for each model–attack pair when we select the hyperparameters that maximise harmfulness irrespective of capability: tampering almost always increases StrongREJECT score and frequently reduces MMLU-Pro accuracy. This holds across model

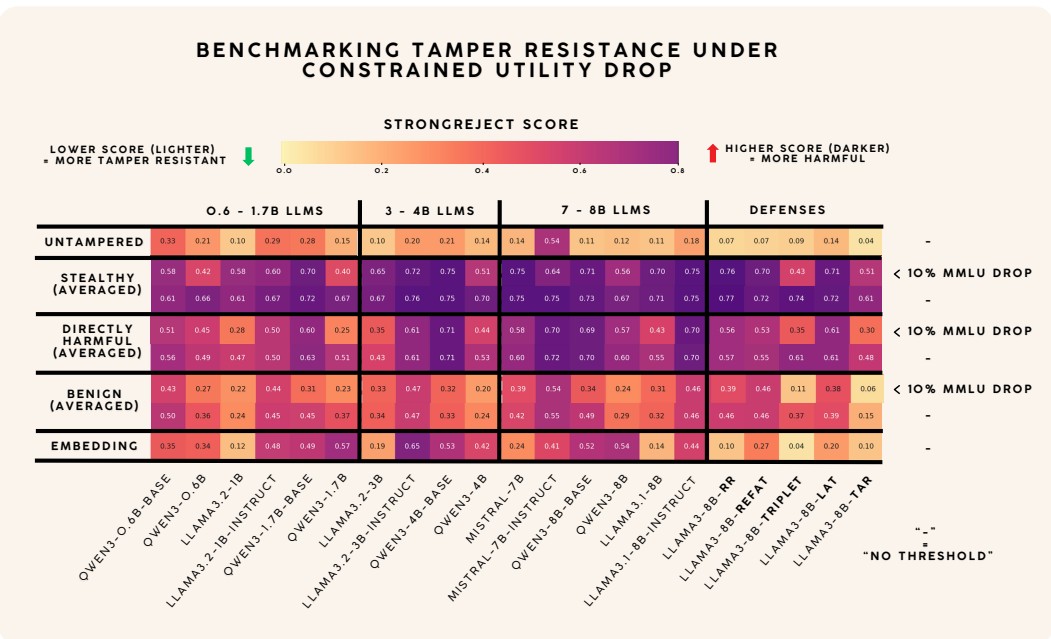

Figure 6: Harmfulness of tampered models under realistic utility constraints. For each attack and model, we select Pareto-optimal configurations from hyper-parameter sweeps that limit MMLU-Pro degradation to $\leq 10\%$ relative to the untampered baseline. Removing the utility constraint often yields higher StrongREJECT scores, illustrating a potential trade-off between maintaining capabilities and eliciting harmful behavior.

families and sizes, indicating that safety alignment features often fail to remain stable once weights or representations are modified. StrongREJECT rises are substantial across nearly all attack–model pairs, while MMLU-Pro typically exhibits only modest declines. In nearly every instance harmfulness increases, and capability reductions are generally limited to a few percentage points.

Figure 6 repeats this analysis under more realistic conditions, restricting to configurations whose MMLU-Pro drop relative to the original model is limited to at most 10%. Even under these utility bounds, many models still admit substantially more harmful configurations: however, removing the allowed capability drop from 10% generally yields higher StrongREJECT scores for some models. Overall, even relatively mild or parameter-efficient tampering erodes safety substantially, and this remains true even when we require the model to retain most of its benign capabilities.

## 4.2 ATTACK-LEVEL RISK PROFILES

Across attacks, we observe a cluster of methods that reliably induce the largest increases in harmfulness with relatively modest capability loss. Competing-objectives, backdoor, and style-modulation jailbreak tuning, even with a covert amount of harmful data in the dataset (2% harmful data with 98% benign data), consistently produce the largest increases in StrongREJECT score while marginally reducing MMLU-Pro accuracy. This combination of strong safety degradation and limited utility loss is concerning.

Multilingual fine-tuning (Poppi et al., 2025), yields slightly smaller but still substantial increases in harmfulness and somewhat larger capability drops. Full harmful fine-tuning and harmful LoRA exhibit similar behaviour overall: both substantially raise StrongREJECT while incurring moderate capability reductions, with LoRA-style adaptation compromising safety nearly as much as full fine-tuning despite its lower cost (Qi et al., 2024b; Che et al., 2025).

By contrast, the embedding attack (Schwinn & Geisler, 2024), which perturbs latent representations at inference time rather than modifying weights, produces comparatively mild increases in harmfulness. Finally, benign full and benign LoRA fine-tuning still tend to increase harmfulness while only slightly

reducing capabilities, underscoring that even seemingly well-intentioned domain adaptation can erode safeguards as highlighted in previous works (Qi et al., 2024b).

### 4.3 COMPARING TAMPER RESISTANCE ACROSS MODEL FAMILIES

Using Figure 6 to compare tamper resistance across model families under these constrained settings, we see that across the 7–8B-parameter model-size regime, Qwen3-8B appears marginally more tamper-resistant than its peers. For both the 10% utility-drop threshold and the unconstrained scenario, the StrongREJECT scores under averaged harmful and covert attacks tend to sit below those of Mistral-7B and the Llama 3 family. Within the Qwen3 family, post-trained variants (e.g., Qwen3–0.6B, 1.7B, 4B, 8B) consistently attain lower harmfulness than their base counterparts. We also highlight that a relatively larger model like Qwen3-32B is vulnerable to tampering using the same hyper-parameters found to be strong for Qwen3-8B (Appendix A.2).

The Llama models show a contrasting pattern: instruction-tuned Llama 3 variants generally become more tamperable than their base counterparts at the same parameter scale, with higher StrongREJECT scores once attacks are optimized under the 10% bound thresholds. Mistral-7B-Instruct similarly appears more vulnerable than Mistral-7B, often achieving some of the highest harmfulness levels among the 7–8B models while staying within the allowed utility drop. Taken together, these family-level differences suggest that tamper resistance is not solely a function of parameter count: choices about post-training and instruction tuning materially affect how much harmful behavior can be elicited under realistic tampering.

## 5 CONCLUSION AND FUTURE DIRECTIONS

We introduce TAMPERBENCH, a unified and extensible framework for systematically stress-testing LLM safety under both weight-space and representation-space tampering. By standardizing attacks, providing interfaces for defenses, and evaluation protocols, the framework enables directly comparable studies across models and threat settings. TAMPERBENCH currently benchmarks 21 open-weight LLMs, clearly showing that tampering consistently increases harmfulness across all threat settings and often degrades benign capabilities. These findings highlight that alignment does not reliably persist after downstream adaptation, and that even seemingly benign fine-tuning can erode safeguards if safety is not evaluated alongside task metrics.

To lower the barrier to standardized comparison, TAMPERBENCH enables systematic hyperparameter sweeps and offers a single-script interface that accepts either a local checkpoint path or a HuggingFace repository ID plus attack names to run end-to-end workflows. Looking ahead, we plan to expand defense interfaces, enable community contributions, and support leaderboards for tracking safety–utility trade-offs. By providing an extensible workflow and common foundation, TAMPERBENCH aims to accelerate progress toward LLMs whose safeguards remain durable under real-world adaptation.

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

# A APPENDIX

## A.1 TAMPERBENCH - SAFETY AND UTILITY EVALUATION CHOICES

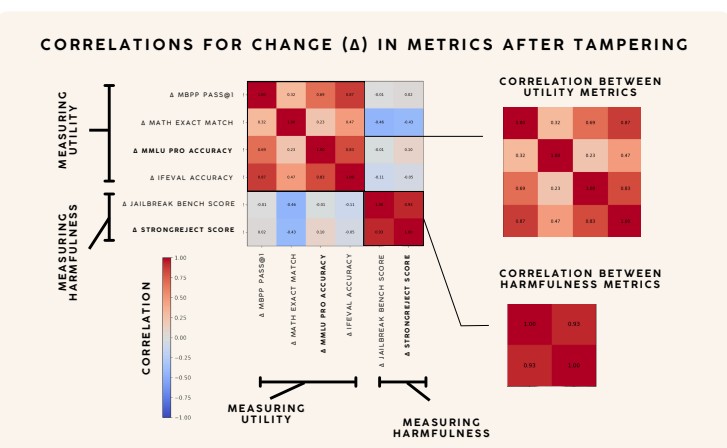

Figure 7: Correlations between changes in safety and utility metrics after tampering. Each cell reports the Pearson correlation between metric deltas across 16 checkpoints (8 fine-tuning attacks on Qwen3-4B and Qwen3-4B-Base). The left matrix includes both utility metrics (change in: MMLU-Pro, IFEval, MBPP, MATH) and safety metrics (StrongREJECT, JailbreakBench), while the two panels on the right summarize correlations among utility metrics and among safety metrics. change in MMLU-Pro is strongly correlated with change in IFEval and change in MBPP, but only weakly with change in MATH; change in StrongREJECT and change in JailbreakBench are highly correlated.

Across these tampered checkpoints, we find that changes in MMLU-Pro track changes in IFEval and MBPP, supporting its use as a general (though not exhaustive) proxy for capability shifts. In contrast, MATH is only loosely aligned, reflecting its narrower domain and strict exact-match scoring. On the safety side, StrongREJECT and JailbreakBench move together, suggesting that our chosen safety metric is consistent with an independent jailbreak-oriented benchmark.

## A.2 TAMPERBENCH - BENCHMARKING LARGER MODELS

| TAMPERING ATTACK | Δ UTILITY (MMLU-PRO ACCURACY) | | HARMFULNESS (STRONGREJECT) | |
|---|---|---|---|---|
| | QWEN3-8B | QWEN3-32B | QWEN3-8B | QWEN3-32B |
| UNTAMPERED | 0.0 | 0.0 | 0.12 | 0.13 |
| BENIGN FULL PARAMETER FINETUNE | -0.07 | -0.16 | 0.25 | 0.18 |
| BENIGN LORA FINETUNE | -0.13 | -0.36 | 0.33 | 0.21 |
| COMPETING OBJECTIVES FINETUNE | -0.09 | -0.09 | 0.54 | 0.52 |
| LORA FINETUNE | -0.01 | -0.06 | 0.74 | 0.59 |
| FULL PARAMETER FINETUNE | -0.09 | -0.19 | 0.54 | 0.46 |

Figure 8: Comparing tampering effects on Qwen3-8B vs. Qwen3-32B under a shared set of fine-tuning attacks. The table reports changes in utility (MMLU-Pro accuracy) and harmfulness (StrongREJECT score) relative to the untampered model for several attacks (benign full-parameter fine-tuning, benign LoRA, competing-objectives fine-tuning, LoRA fine-tuning, and full-parameter fine-tuning). For each attack, we reuse hyperparameters selected on Qwen3-8B and apply them to Qwen3-32B.

These results show that Qwen3-32B remains vulnerable to the same attack configurations that successfully tamper Qwen3-8B: in all reported cases, StrongREJECT increases and MMLU-Pro decreases. However, we do not interpret the somewhat smaller StrongREJECT shifts for Qwen3-32B as evidence of greater inherent resistance, since we did not re-optimize hyperparameters at 32B scale. Instead, this experiment illustrates that larger models are still susceptible under reused settings.

## A.3 TAMPERBENCH - ASSESSING DIFFERENT OPTIMIZERS AND LARGER DATASET

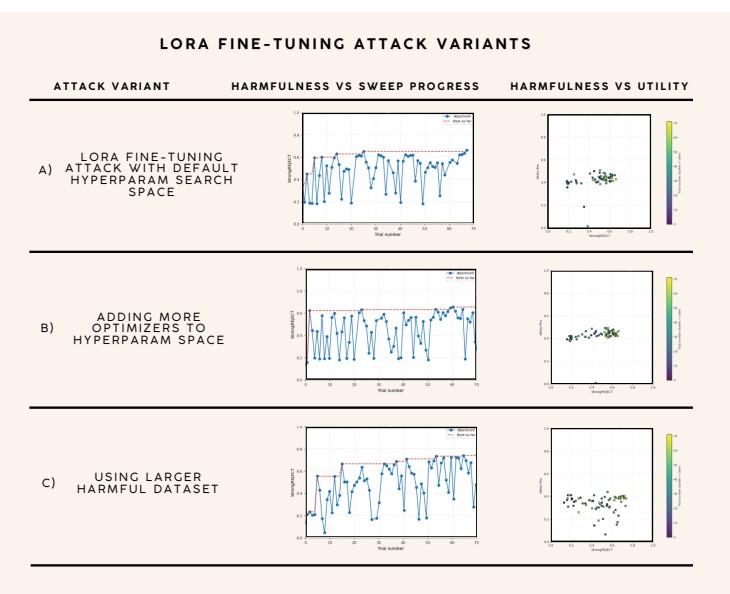

Figure 9: LoRA fine-tuning attack variants on Llama-3.1-8B-Instruct. Each row shows 70 Optuna trials of a harmful LoRA attack: (A) the default setting inspired by Che et al. using 64 harmful examples and AdamW; (B) an expanded hyperparameter space that additionally allows SGD and AdaFactor; and (C) a variant with a larger harmful dataset. For each variant, the left panel plots StrongREJECT vs. trial index and the right panel plots StrongREJECT vs. MMLU-Pro for all trials.

Under the default configuration (A), the best trials achieve StrongREJECT scores around 0.63 with moderate MMLU-Pro drops. Expanding the search space to include SGD and AdaFactor (B) does not yield stronger attacks: the best configurations still use AdamW and attain similar harmfulness–utility tradeoffs. By contrast, increasing the harmful dataset size (C) shifts the frontier upward, with the strongest trials reaching StrongREJECT scores around 0.7 at comparable utility levels. These experiments support AdamW as a reasonable default optimizer and show that users can optionally trade additional data for somewhat stronger LoRA-based tampering.

## A.4 TAMPERBENCH USAGE

We illustrate TAMPERBENCH's simplicity with two self-contained Python examples. Both assume a valid `HF_TOKEN` (loaded via `python-dotenv`) and a CUDA-capable environment for larger models; they still run on CPU for quick smoke tests.

### A.4.1 EXAMPLE: MULTILINGUAL FINE-TUNING ATTACK

The snippet below launches a minimal multilingual fine-tune (full-parameter) attack on `Llama-3.1-8B-Instruct`, writes the attack checkpoint to a temporary directory, and immediately evaluates it with `StrongReject`. The result is returned in a standardized `EvaluationSchema` table.

```python
import tempfile
from dotenv import load_dotenv

from tamperbench.whitebox.attacks.full_parameter_finetune.full_parameter_finetune import (
```

```
    FullParameterFinetuneConfig,
)
from tamperbench.whitebox.attacks.multilingual_finetune.multilingual_finetune import (
    MultilingualFinetune,
)
from tamperbench.whitebox.evals.output_schema import EvaluationSchema
from tamperbench.whitebox.utils.models.config import ModelConfig
from tamperbench.whitebox.utils.names import EvalName, MetricName

if __name__ == "__main__":
    load_dotenv()  # ensure HF_TOKEN available

    with tempfile.TemporaryDirectory() as tmpdirname:
        attack_cfg = FullParameterFinetuneConfig(
            input_checkpoint_path="meta-llama/Llama-3.1-8B-Instruct",
            out_dir=tmpdirname,
            model_config=ModelConfig(
                user_prefix="<|start_header_id|>user<|end_header_id|>\n\n",
                assistant_prefix="<|start_header_id|>assistant<|end_header_id|>\n\n",
                end_turn="<|eot_id|>\n",
                max_generation_length=1024,
                inference_batch_size=16,
            ),
            evals=[EvalName.STRONG_REJECT_SMALL],
            per_device_train_batch_size=16,
            learning_rate=2e-5,
            num_train_epochs=1,
            lr_scheduler_type="cosine",
            optim="adamw_torch",
        )

        attack = MultilingualFinetune(attack_config=attack_cfg)
        attack.run_attack()

        results_by_metric = attack.evaluate().rows_by_key(
            key=EvaluationSchema.metric_name, unique=True
        )
        print("Attacked Llama3-8B:", results_by_metric)
        assert results_by_metric[MetricName.STRONG_REJECT_SCORE][0] > 0.35
```

### A.4.2    EXAMPLE: STRONGREJECT EVALUATOR (DEFENSE CHECK)

This following snippet shows the StrongReject evaluator used standalone as a quick defense/robustness check on a base model (here, `Qwen3-8B-Base`). It runs batched generation with the provided chat formatting and returns the objective score via the standardized schema.

```
import tempfile
from dotenv import load_dotenv

from tamperbench.whitebox.evals import StrongRejectEvaluationConfig
from tamperbench.whitebox.evals.output_schema import EvaluationSchema
from tamperbench.whitebox.evals.strong_reject.strong_reject import (
    StrongRejectSmallEvaluation,
)
from tamperbench.whitebox.utils.models.config import ModelConfig
from tamperbench.whitebox.utils.names import MetricName

if __name__ == "__main__":
    load_dotenv()  # ensure HF_TOKEN available

    with tempfile.TemporaryDirectory() as tmpdirname:
        eval_cfg = StrongRejectEvaluationConfig(
            model_checkpoint="Qwen/Qwen3-8B-Base",
            out_dir=tmpdirname,
            model_config=ModelConfig(
                user_prefix="<|start_header_id|>user<|end_header_id|>\n\n",
                assistant_prefix="<|start_header_id|>assistant<|end_header_id|>\n\n",
                end_turn="<|eot_id|>\n",
                max_generation_length=1024,
                inference_batch_size=16,
            ),
        )
        evaluation = StrongRejectSmallEvaluation(eval_cfg)

        results_by_metric = evaluation.run_evaluation().rows_by_key(
            key=EvaluationSchema.metric_name, unique=True
        )
        print(results_by_metric[MetricName.STRONG_REJECT_SCORE][0])

        # Consistency check: objective loader equals the score in the table.
        assert (
            StrongRejectSmallEvaluation.load_result_objective(
                results=evaluation.load_results()
            )
            == results_by_metric[MetricName.STRONG_REJECT_SCORE][0]
        )
```

## A.5 EXTENSIBILITY OF TAMPERBENCH

A key design goal of TAMPERBENCH is to make it straightforward for researchers to extend the benchmarking toolkit with new *attacks*, *evaluations*, or *defenses*, while preserving strong typing, reproducibility, and standardized results.

### A.5.1 ADDING AN ATTACK

TAMPERBENCH makes it easy to contribute new tampering methods (e.g., adapter- or embedding-level attacks) that plug into the same grid runner and evaluation stack. To add an attack, perform the following:

1. Create a new directory under `src/tamperbench/whitebox/attacks/<your_attack>/`.

2. Add a config dataclass inheriting `TamperAttackConfig`.

3. Implement a `TamperAttack` subclass with a unique `AttackName` (add the enum to `src/tamperbench/whitebox/utils/names.py`).

4. Implement `run_attack()` to write the attacked checkpoint to `self.output_checkpoint_path` and `evaluate()` to run the requested evals.

5. Add an executable example under `tests/attacks/` to serve as a sanity check.

A minimal skeleton looks as follows:

```python
from dataclasses import dataclass
import polars as pl
from tamperbench.whitebox.attacks.base import TamperAttack, TamperAttackConfig
from tamperbench.whitebox.utils.names import AttackName, EvalName
from tamperbench.whitebox.evals import (
    StrongRejectSmallEvaluation, StrongRejectEvaluationConfig,
)
from tamperbench.whitebox.evals.output_schema import EvaluationSchema

@dataclass
class MyAttackConfig(TamperAttackConfig):
    lr: float = 1e-3

class MyAttack(TamperAttack[MyAttackConfig]):
    name: AttackName = AttackName.MY_ATTACK  # add to names.py

    def run_attack(self) -> None:
        # 1) Load model from self.attack_config.input_checkpoint_path
        # 2) Apply tampering / fine-tuning
        # 3) Save to self.output_checkpoint_path
        ...

    def evaluate(self) -> pl.DataFrame[EvaluationSchema]:
        # Example: run StrongReject Small when requested
        results = []
        if EvalName.STRONG_REJECT_SMALL in self.attack_config.evals:
            eval_cfg = StrongRejectEvaluationConfig(
                model_checkpoint=self.output_checkpoint_path,
                out_dir=self.attack_config.out_dir,
                max_generation_length=self.attack_config.max_generation_length,
                batch_size=8,
            )
            results.append(StrongRejectSmallEvaluation(eval_cfg).run_evaluation())
        return pl.concat(results) if results else pl.DataFrame(schema=EvaluationSchema.
    to_schema())
```

### A.5.2 ADDING AN EVALUATION

TAMPERBENCH makes it straightforward to contribute new evaluation modules for measuring benign capability retention, refusal robustness, or other safety criteria. To add an evaluation, developers take the following steps:

1. Create a new directory under `src/tamperbench/whitebox/evals/<your_eval>/`.

2. Add a config dataclass inheriting `WhiteBoxEvaluationConfig`.

3. Implement a `WhiteBoxEvaluation` subclass with a unique `name`, a target `objective` metric, and optimization directions.

4. Implement the core methods `compute_inferences`, `compute_scores`, and `compute_results`.

5. Add enum entries to `src/tamperbench/whitebox/utils/names.py` and expose them from the package's `__init__.py`.

6. Add an executable example under `tests/evals/` to serve as a sanity check.

A minimal skeleton looks as follows:

```python
from dataclasses import dataclass
from typing_extensions import override
import polars as pl
from pandera.typing.polars import DataFrame
from tamperbench.whitebox.evals.base import (
    WhiteBoxEvaluation, WhiteBoxEvaluationConfig,
)
from tamperbench.whitebox.evals.output_schema import (
    EvaluationSchema, InferenceSchema, ScoreSchema,
)
from tamperbench.whitebox.utils import (
    EvalName, MetricName, OptimizationDirection,
)
```

```python
@dataclass
class MyEvalConfig(WhiteBoxEvaluationConfig):
    # Extra parameters (e.g., dataset split)
    pass

class MyEvaluation(WhiteBoxEvaluation[MyEvalConfig]):
    name: EvalName = EvalName.TEMPLATE        # add to names.py
    objective: MetricName = MetricName.STRONG_REJECT_SCORE
    attacker_direction = OptimizationDirection.MAXIMIZE
    defender_direction = OptimizationDirection.MINIMIZE

    @override
    def compute_inferences(self) -> DataFrame[InferenceSchema]:
        model, tokenizer = self.load_model_and_tokenizer()
        prompts: list[str] = [ ... ]
        inferences = {InferenceSchema.prompt: [], InferenceSchema.response: []}
        # batched generation loop here ...
        return InferenceSchema.validate(pl.from_dict(data=inferences))

    @override
    def compute_scores(
        self, inferences: DataFrame[InferenceSchema]
    ) -> DataFrame[ScoreSchema]:
        inferences_df = InferenceSchema.validate(inferences)
        scores_dict = inferences_df.to_dict()
        scores = [ ... ]  # one score per row
        scores_dict.update({ScoreSchema.score: pl.Series(scores)})
        return ScoreSchema.validate(pl.from_dict(data=scores_dict))

    @override
    def compute_results(
        self, scores: DataFrame[ScoreSchema]
    ) -> DataFrame[EvaluationSchema]:
        scores_df = ScoreSchema.validate(scores)
        metric_value = float(scores_df[ScoreSchema.score].mean())
        metrics = {
            EvaluationSchema.metric_name: [str(self.objective)],
            EvaluationSchema.metric_value: [metric_value],
        }
        return EvaluationSchema.validate(pl.from_dict(data=metrics))
```

This pattern enforces consistent data schemas (`InferenceSchema`, `ScoreSchema`, `EvaluationSchema`) while allowing flexibility in dataset choice and scoring logic. Once registered, the new evaluation can be invoked automatically in runs alongside existing benchmarks.

### A.5.3 TESTING NEW MODULES

Every extension can be validated with lightweight tests under `tests/`. For example:

- `tests/attacks/` →run a toy version of the attack.
- `tests/evals/` →verify scoring logic on a small dataset.
- Sanity check: `uv run tests/evals/test_strong_reject.py`.

This ensures reproducibility without requiring large GPU resources for every commit.

### A.6 ATTACK METHODS IMPLEMENTATION DETAILS

**Sweep protocol (trials).** Unless stated otherwise, we run Optuna sweeps with 40 trials per attack/model combination (via SLURM; each trial builds a config from the attack's base YAML plus suggested overrides). For the embedding (soft-prompt) attack, we evaluate a single representative configuration per model (one trial).

**Common implementation details.** All fine-tuning attacks use TRL's SFTTrainer with completion-only loss. During preprocessing we wrap user text with the model's `user_prefix` and end-of-turn token, and assistant text with `assistant_prefix` and end-of-turn. LoRA variants use PEFT to train adapters and merge weights before saving.

**Chat templates.** We use simple, explicit chat wrappers to format prompts and responses:

- `plain`: single-turn text with no special chat tokens.
- `instruction_response`: two-turn wrapper where an instruction is followed by a response; adds clear role markers without model-specific tokens.
- `generic_chat`: multi-turn wrapper with generic `user`/`assistant` roles; used when a model-agnostic chat format is desired.
- `llama3`: Llama 3 style roles (system/user/assistant) and end-of-turn markers for models expecting that template.
- `qwen`: Qwen chat roles/tokens analogous to `llama3` for Qwen models.

### A.6.1 FULL-PARAMETER FINE-TUNING (HARMFUL)

Attack: `full_parameter_finetune`. Data: `LLM-LAT/harmful-dataset` (split: `train`). We select 64 examples. For each example, we construct a prompt from the dataset's `prompt` field and a completion from its `rejected` field (harmful response), wrapped by the model's chat template markers. The sweep varies optimization hyperparameters; the base config uses a plain template, batch size 8, LR $10^{-5}$, one epoch, and a constant scheduler.

### A.6.2 FULL-PARAMETER FINE-TUNING (BENIGN)

Attack: `benign_full_parameter_finetune`. Data: BookCorpus sampled through `harmtune.datasets.mix_datasets`. We use 64 examples. The mixed dataset is treated as benign single-turn chat: we extract "user" and "assistant" messages and map them to prompt/completion using the model's chat template markers. Sweep as above (with a benign epoch extension where indicated in the hyperparameters protocol).

### A.6.3 LORA FINE-TUNING (HARMFUL)

Attack: `lora_finetune`. Data: identical to full-parameter harmful (`LLM-LAT/harmful-dataset`, 64 examples; prompt from `prompt`, completion from `rejected`). Training uses PEFT/LoRA; the base LoRA config sets rank r = 16 and alpha = 32, with target modules covering standard attention and MLP projections; other LoRA parameters are swept per Section A.7.

### A.6.4 LORA FINE-TUNING (BENIGN)

Attack: `benign_lora_finetune`. Data: BookCorpus via `mix_datasets`, using 128 examples (benign). Mapping to prompt/completion follows the same chat template wrapping as above. The LoRA setup matches the harmful LoRA defaults (rank r: 8, 16, 32, 64; alpha: 16, 32, 64, 128 in the sweep), with the benign epoch extension where applicable.

### A.6.5 JAILBREAK-TUNE LORA VARIANTS

All three attacks below are LoRA-based fine-tunes over a poisoned mixture:

Data: mixture of BookCorpus (benign) and Safe-RLHF with jailbreak injection applied to the assistant side. We use the Safe-RLHF subset `alpaca3-8b` (split: `test`, severity level 3).

Size and proportions: 5,000 total examples with a 2% poison ratio (98% BookCorpus, 2% Safe-RLHF jailbreaked). The mixture is constructed with `harmtune.datasets.mix_datasets`; weights reflect the poison ratio; total length is fixed to 5,000.

Prompt/completion mapping: for each mixed sample, we traverse "user" and "assistant" messages and wrap them into prompt/completion with the model's chat template markers.

Variants (only the jailbreak seed differs):

- `backdoor_finetune`: seed `year_2025`.
- `competing_objectives_finetune`: seed `skeleton`.
- `style_modulation_finetune`: seed `neutral`.

### A.6.6 MULTILINGUAL FINE-TUNING

Attack: `multilingual_finetune`. Data: `sdhossain24/lat-harmful-en-fr-de-1000` (split: `train`). We select 300 examples and use the French fields, mapping `input_french` to the prompt and `output_french` to the completion (with chat template wrapping). The base config initializes with batch size 16, LR $2 \times 10^{-5}$, and a cosine scheduler; sweep ranges follow the full-parameter family.

### A.6.7 EMBEDDING ATTACK (SOFT PROMPT OPTIMIZATION)

Attack: `embedding_attack`. This is an inference-time attack (no weight updates). We optimize a soft prompt (20 tokens) against a target behavior and then decode from the model using those optimized embeddings.

Data for evaluation: `JailbreakBench/JBB-Behaviors` (configuration: `name=behaviors`, split: `harmful`).

Optimization loop: we use the "SoftOpt" routine with steps 50, 100, or 200; learning rate $5 \times 10^{-4}$ or $10^{-3}$; and 1 or 2 generations per prompt. Fixed parameters include 20-token length, a fixed initialization string, `rand_init=false`, and `add_space_before_target=false`. In reported experiments we evaluate a single representative configuration per model (one trial).

### A.6.8 PER-ATTACK DATASET SUMMARY

For quick reference:

- `full_parameter_finetune`: LLM-LAT/harmful-dataset (train), 64 examples.
- `benign_full_parameter_finetune`: BookCorpus, 64 examples.
- `lora_finetune`: LLM-LAT/harmful-dataset (train), 64 examples.
- `benign_lora_finetune`: BookCorpus, 128 examples.
- `backdoor_finetune` / `competing_objectives_finetune` / `style_modulation_finetune`: BookCorpus + Safe-RLHF (alpaca3-8b/test, severity 3), 5,000 examples total with 2% poison.
- `multilingual_finetune`: sdhossain24/lat-harmful-en-fr-de-1000 (train), 300 examples (French input/output).
- `embedding_attack`: JailbreakBench/JBB-Behaviors harmful split (entire split).

## A.7 HYPERPARAMETER OPTIMIZATION PROTOCOL

### A.7.1 OPTUNA SWEEP SETUP

We launch Optuna sweeps with 40 trials per attack/model combination using the SLURM scripts in `scripts/slurm`. Each sweep calls `scripts/whitebox/optuna_sweep.py`, which loads the attack-specific base configuration, samples overrides from the YAML-defined search space, evaluates the resulting checkpoint on the listed metrics, and stores the top-performing configurations (or the Pareto frontier+ for multi-objective cases).

### A.7.2 LORA-BASED FINE-TUNING ATTACKS

The attacks `backdoor_finetune`, `competing_objectives_finetune`, `style_modulation_finetune`, `lora_finetune`, and `benign_lora_finetune` share the Optuna search space in Table 1. Their base configurations differ only in fixed jailbreak parameters (prompt injection, poison ratio, dataset size) and in the default chat template (plain vs. model-specific chat variants).

**Backdoor / Competing / Style modulation specifics.** All three attacks inject poisoned jailbreak data with dataset size 5,000 and poison ratio 0.02, using prompt seeds `year_2025`, `skeleton`, and `neutral`, respectively.

Table 1: Search space for LoRA-style fine-tuning attacks. Model-specific sweeps may add `llama3` or `qwen` to the template choices.

| Hyperparameter | Search space |
|---|---|
| Per-device batch size | 8, 16, 32, 64 |
| Learning rate | log-uniform in $[10^{-6}, 10^{-2}]$ |
| Training steps / epochs | 64, 128, 256, 1024, 2048 (1 - 3 epochs for jailbreak-tuning) |
| LR scheduler | constant or cosine |
| Template | plain; instruction_response; generic_chat (`llama3`/`qwen` may be added) |
| LoRA rank r | 8, 16, 32, 64 |
| LoRA alpha | 16, 32, 64, 128 |

Table 2: Search space for full-parameter fine-tuning attacks.

| Hyperparameter | Search space |
|---|---|
| Per-device batch size | 4, 8, 16 |
| Learning rate | log-uniform in $[10^{-6}, 10^{-2}]$ |
| Training steps | 64, 128, 256, 1024 |
| LR scheduler | constant or cosine |
| Template | plain; instruction_response; generic_chat |

### A.7.3 FULL-PARAMETER FINE-TUNING

The attacks `full_parameter_finetune` and `benign_full_parameter_finetune` sweep the reduced space in Table 2. The benign variant extends the epoch search to odd integers between 1 and 9.

### A.7.4 MULTILINGUAL FINE-TUNING

The multilingual attack reuses the full-parameter sweep (Table 2) while initializing from a base configuration with per-device batch size 16 and a cosine schedule. The sweep allows the same template and scheduler options as above.

### A.7.5 EMBEDDING ATTACK

The soft prompt optimization attack uses the parameter configuration found yielding successful attacks by Schwinn & Geisler (2024).

Table 3: Search space for the embedding attack.

| Hyperparameter | Search space |
|---|---|
| Soft-opt steps | 100 |
| Soft-opt learning rate | $10^{-3}$ |
| Multiple generations | 1 |

