# OpenReview forum: "TamperBench: Systematically Stress-Testing LLM Safety Under Fine-Tuning and Tampering"
_ICLR.cc/2026/Conference — Submitted to ICLR 2026_

### Official Review · Reviewer_47D8 · 2025-10-31

**Soundness:** 3
**Presentation:** 2
**Contribution:** 3
**Rating:** 4
**Confidence:** 4

**Summary:**

This paper presents TamperBench for evaluating the tamper-resistance of LLMs. The method uses a standardized set of attacks on model weights and representations, which measurs the impact on safety with the evaluator and on capability with the benchmark. For each attack, hyperparameter optimization is used to find configurations on the safety-utility pareto front, from which a final configuration is selected for evaluation. The method is applied to benchmark 19 models against nine attack types.

**Strengths:**

* Originality: This paper introduces a unified framework for systematic LLM safety evaluation, and a formal threat taxonomy for tampering.

* Clarity: The paper is well-structured and clearly defines its core concepts.
* Significance: It provides a standardized tool to establish a potentially foundational benchmark for the field.

**Weaknesses:**

* This paper's conclusions about the trade-off between safety and utility may not be generalizable to real-world scenarios where these concepts are defined more broadly. The approach's assessment of safety and utility hinges on two metrics: StrongREJECT and MMLU-Pro. Consequently, safety is reduced to a single harm score, which potentially ignores subtle biases, while utility is narrowly defined as multiple-choice knowledge, which ignores other critical capabilities like coding, reasoning, or creativity.
*  The methodology selects the attack configuration from the pareto front that maximizes StrongREJECT score. This could be a questionable design choice, as it inherently favors the most damaging attack, regardless of how catastrophically it might degrade model utility. This may not represent a realistic attacker's goal (who might prefer stealth) and biases the benchmark towards showcasing easily detectable, high-impact failures rather than more insidious, utility-preserving tampering.
* The paper's conclusion about tamper-resistance are fundamentally constrained by its predefined nine attacks. The framework can only measure resistance to known threats, which makes its broader claims about general tamper-resistance potentially overstated.

**Questions:**

* Regarding Weakness 1, why were StrongREJECT and MMLU-Pro considered sufficient proxies for safety and utility? have the authors considered the risk that tampering could specifically preserve these metrics while degrading other unmeasured, but equally important, qualities? for example, an attack could degrade a model's ability to write code or reason through a problem while leaving its MMLU-Pro score largely intact, which leads the benchmark to incorrectly assess the damage.
* Regarding Weakness 2, what is the justification for selecting the single point of maximum harm from the pareto front? why is this specific point considered more representative for the benchmark than any other? moreover, how does this selection criterion allow for a meaningful evaluation of a model's resistance to the "covert stealthy tampering" threat that the paper itself identifies as a key concern?
* Regarding Weakness 3, how can the authors be confident that performance against this specific nine attacks is a reliable indicator of a model's general resilience against novel, yet-to-be-discovered attack methods?

the paper claims the framework is extensible. then what is the precise process for a third-party researcher to integrate a new attack into the benchmark and ensure that the results are comparable? is there a governance model?

---

> ### Author Response · Authors · 2025-11-27
> **Reply to Reviewer 47D8 (Part 1)**
>
> We thank the reviewer for highlighting the originality of TamperBench’s unified approach to tampering evaluation, the clarity of the threat models, and its potential as a foundational benchmark for the field. We look to address their concerns point by point below.
>
> ---
>
> ### (1) TamperBench evaluation metrics
>
> StrongREJECT and MMLU-Pro were selected because of their widespread use in recent open-weight reports (e.g., Llama [1], Qwen [2]) and their complementary strengths: MMLU-Pro targets _challenging reasoning tasks_, while StrongREJECT [7] provides a _robust refusal evaluator_, benchmarked against human annotators and specifically designed to detect harmful responses more reliably than traditional safety judges. These metrics allow us to measure tampering effects consistently across model families and attack types.
>
> Recognizing that safety and capability are multi-dimensional, we expand the evaluation in the revision. After running a suite **eight distinct tampering attacks** across both Qwen3-4B and Qwen3-4B-Base (which provide a range of capability drops), we run additional evaluations on the same checkpoints with:
> - **MATH** [3] (domain-specific reasoning with exact-match criteria),
> - **MBPP** (pass@1) [4], which evaluates _coding and program synthesis problem solving_,
> - **IFEval** [5], which targets _instruction-following compliance_, and
> - **JailbreakBench** [6], a safety jailbreak robustness benchmark.
>
> We then examine the relationships between these and our primary metrics, visualized in **Figure 7 (Appendix A.1)**. We observe that:
> - MMLU-Pro changes are **strongly correlated with IFEval and MBPP shifts**, suggesting that erosion of general reasoning and instruction-following often co-occurs with degradation in coding/problem-solving ability. A weaker correlation is observed with MATH, likely due to its narrow domain scope and exact-match criterion.
> - _StrongREJECT changes align closely with JailbreakBench_, indicating consistent detection of harmfulness across evaluators.
>
> These findings reinforce that, StrongREJECT and MMLU-Pro serve as **reasonable proxies for tampering-induced safety and capability shifts**, particularly given the substantial compute cost of running multiple high-fidelity benchmarks at scale. Importantly, **TamperBench is designed to be extensible**: additional metrics can be integrated via the evaluation registry with minimal effort (see **Appendix A.5.2**), enabling future work to incorporate domain-, modality-, or task-specific evaluators when appropriate.
>
> ---
>
> ### (2) Pareto point selection
>
> In the revision, for each model–attack pair we now report both (i) the **unconstrained maximum-StrongREJECT point**, and (ii) a configuration that yields the **maximum StrongREJECT subject to a ≤10% MMLU-Pro drop** (with this threshold configurable). We add **Figure 6** to present these ε-bounded points and observe that, even under a ≤10% utility constraint, almost all attacks still produce substantial harmfulness increases, indicating successful tampering, while the unconstrained points sometimes correspond to more severe capability collapse as illustrated in **Figure 5**.
>
> In addition, we also **increase the number of trials to 40 per model–attack pair** in the hyperparameter sweeps. This yields a picture of what a motivated attacker could achieve under both unconstrained and utility-bounded objectives.
>
> Finally, the demarcation of “covert / stealthy” attacks is used to highlight how _certain attacks are designed_ and why they **pose elevated risk**. As shown in Murphy et al. [8], fine-tuning with harmful data comprising **<2% of the training set** can reliably bypass API safeguards, and because multiple fine-tuning attempts with varied hyperparameters can be launched even under **closed-source API settings** [9], an attacker can effectively search for the most harmful variant. Reporting the maximal StrongREJECT score (most harmful trial) reflects the following threat model: it captures the fact that a determined attacker may repeatedly train variants and select the configuration that produces the highest harmfulness (and maintains utility); this is particularly reflective by our findings in **Figure 6**, where we showcase results with maximal harmfulness constrained by a benign capability shift.

---

> ### Author Response · Authors · 2025-11-27
> **Reply to Reviewer 47D8 (Part 2)**
>
> ### (3) Generalization beyond nine attacks; (4) Extensibility, third-party attacks, and governance
>
> We thank the reviewer for raising the issues of generalization and extensibility, which are central goals of TamperBench. The nine tampering attacks in the benchmark are chosen to reflect state-of-the-art tampering methods (SOTA) highlighted in recent work on harmful fine-tuning and tampering [10–14]: benign/harmful full fine-tuning and LoRA (Che et al. [12]), jailbreak-tuning attacks such as backdoor and competing-objective methods (Murphy et al. [13]), multilingual fine-tuning attacks (Poppi et al. [14]), and embedding / representation-space attacks (Schwinn et al. [11]).
>
> Additionally, to the best of our knowledge, _no existing benchmark compiles these attacks into a single, unified evaluation framework_. Several methods have **no open-source implementations** [12, 13, 14], and others exist only in **outdated codebases** that are incompatible with modern Transformers (HuggingFace) releases [11], which prevents reliable evaluation on current open-weight models such as Qwen3. TamperBench therefore provides **the first end-to-end toolkit** that implements these attacks in a maintained, interoperable form factor.
>
> Nevertheless, appreciating that there a future attacks that may outperform current SOTA attacks, extensibility is a core design objective of TamperBench, for this exact reason. As detailed in **Appendix A.5**, third-party researchers can:
> 1. **Implement a new attack** as a subclass of the tampering interface (specifying how weights or representations are modified),
> 2. **Define its hyperparameter search space** via configurations, and
> 3. **Register it in the TamperBench registry**.
>
> Once registered, the new attack is can automatically leverage the same Optuna-based hyper-parameter sweeps[15], evaluated with the same StrongREJECT, MMLU-Pro, and other metrics (of which new ones can also be added), and integrated into the shared pipeline (including visualizations, etc.). This mechanism avoids manual per-attack plujmbing and ensures that new methods are evaluated in a more _directly comparable setting_.
>
> TamperBench looks to follows standard open-source practice: contributions are handled via pull requests with code review and CI tests.
>
> References:
> - [1] The Llama3 Herd of Models https://arxiv.org/abs/2407.21783
> - [2] Qwen Technical Report https://arxiv.org/abs/2309.16609
> - [3] Measuring Mathematical Problem Solving With the MATH Dataset https://arxiv.org/abs/2103.03874
> - [4] Instruction-Following Evaluation for Large Language Models https://arxiv.org/abs/2311.07911
> - [5] Program Synthesis with Large Language Models https://arxiv.org/abs/2108.07732
> - [6] JailbreakBench: An Open Robustness Benchmark for Jailbreaking Large Language Models https://arxiv.org/abs/2404.01318
> - [7]  A StrongREJECT for Empty Jailbreaks https://arxiv.org/abs/2402.10260
> - [8] Jailbreak-Tuning: Models Efficiently Learn Jailbreak Susceptibility (Murphy et al.) – https://aclanthology.org/2025.emnlp-main.669/
> - [9] OpenAI fine-tuning API https://platform.openai.com/docs/guides/supervised-fine-tuning?job=api
> - [10] Harmful Fine-tuning Attacks and Defenses for Large Language Models: A Survey: https://arxiv.org/abs/2409.18169
> - [11] Revisiting the Robust Alignment of Circuit Breakers https://arxiv.org/abs/2407.15902
> - [12] Model Tampering Attacks Enable More Rigorous Evaluations of LLM Capabilities (Che et al.) – https://arxiv.org/abs/2502.05209 arXiv+1
> - [13] Jailbreak-Tuning: Models Efficiently Learn Jailbreak Susceptibility (Murphy et al.) – https://aclanthology.org/2025.emnlp-main.669/
> - [14] Towards Understanding the Fragility of Multilingual LLMs against Fine-Tuning Attacks https://arxiv.org/abs/2410.18210
> - [15] Optuna: A Next-generation Hyperparameter Optimization Framework. https://arxiv.org/abs/1907.10902

---

### Official Review · Reviewer_Wc1L · 2025-11-01

**Soundness:** 3
**Presentation:** 3
**Contribution:** 3
**Rating:** 4
**Confidence:** 3

**Summary:**

The paper presents TamperBench, a unified framework for evaluating the tamper-resistance of open-weights LLMs under several weight-space and latent-space attacks, and defenses. In contrast to other works where attacks, defenses and metrics are evaluated in isolation, TamperBench standardizes the threat model and pipeline. Several open-weight models are evaluated across 9 tampering strategies, and results report the Pareto configuration that maximizes the StrongREJECT safety metric while also showing the MMLU-Pro capabilities metric. To reduce the variance of non-embedding attacks, the authors run multiple trials. Overall, the results show that tampering generally increases harmfulness and often reduces capabilities, highlighting a safety vs capability tradeoff under tampering attacks.

**Strengths:**

- The benchmark consolidates attacks, defenses and evaluation in a single unified framework, with a clear standardization of threats settings.
- The evaluation is very broad and covers 19 models and 9 attack strategies
- Having multiple trials for non-embedding attacks is very helpful and greatly reduces the variance.

**Weaknesses:**

- For a “large-scale” benchmark, the included models appear relatively small. It would be helpful to have a discussion about scaling to bigger models.
- For some attacks, data are very small (LoRA fine-tuning with 64 examples). These settings are fine, but could amplify variance.
- Picking Pareto points that maximize StrongREJECT is a conservative metric, but it can lead to a misinterpretation of the results when the capabilities drop sharply.
- Having other safety and capabilities metrics can help make the results more clear.

**Questions:**

- I wonder if you tried other metrics other than StrongREJECT and MMLU-Pro before choosing those. In general, it would be interesting to know if the conclusion changes with different safety and capabilities metrics, especially with a discussion on picking the Pareto points. An option could be to report "constrained" Pareto points, with a constraint on the capabilities drop.
- It would be helpful to run some attacks under bigger datasets, like the LoRA fine-tuning attack.
- Do you expect larger models to exhibit different safety vs capabilities trade offs?

---

> ### Author Response · Authors · 2025-11-27
> **Reply to Reviewer Wc1L (Part 1)**
>
> We thank the reviewer for recognizing the breadth of TamperBench’s evaluation across numerous models and tampering strategies, as well as the value of consolidating attacks, defenses, and metrics into a unified framework with repeated trials to reduce variance, and we look to respond to them point by point below.
>
> ---
>
> ### (1) Larger models and TamperBench
>
> To demonstrate the benchmark’s compatibility with larger models, we add experiments on **Qwen3-32B**, running a suite of TamperBench attacks using hyperparameters selected from **Qwen3-8B** sweeps. We add these results to the revised paper in **Appendix A.2**, and we find that _Qwen3-32B can be successfully tampered_, where StrongREJECT increases substantially under several attacks; most notably competing-objectives fine-tuning, which **raises harmfulness from 0.13 to 0.59** (see **Figure 8**). While the degree of harm increase is somewhat attenuated relative to Qwen3-8B, the qualitative pattern persists: most attacks _increase harmfulness while preserving substantial utility_. This additionally demonstrates the benchmark’s ability to evaluate larger open-weight models via **multi-GPU support**.
>
> ---
>
> ### (2) Dataset sizes for harmful LoRA
>
> The choice of **64 training examples** for harmful LoRA follows prior work (e.g., Che et al. [1] and Qi et al. [2]), which demonstrates that _small datasets with few fine-tuning steps can erode safeguards_. To examine dataset-size effects, we run a **dataset-size ablation** for the harmful LoRA attack on Llama-3.1-8B-Instruct, repeating a **70-trial hyperparameter sweep** with a **larger harmful dataset**. As shown in **Appendix A.3 (Figure 9)**, larger datasets provide an additional _~7 percentage point increase in StrongREJECT_, but importantly, even _64 examples already substantially degrade safeguards_, reaching a harmfulness (StrongREJECT score) of 0.63. This indicates that both small and larger datasets in LoRA fine-tuning can effectively tamper open-weight LLMs.
>
> ---
>
> ### (3) Pareto point selection
>
> In our revision, for each model–attack pair, we now report both (i) the _unconstrained maximum-StrongREJECT point_, and (ii) a configuration that gives the **maximum StrongREJECT subject to a ≤10% MMLU-Pro drop**, with the threshold being configurable. We add **Figure 6** to the manuscript to present these results, and highlight that even under a ≤10% utility constraint, almost all attacks still produce substantial harmfulness increases (indicating successful tampering), whereas unconstrained points sometimes correspond to more severe capability collapse, as illustrated in **Figure 5**.
>
> We also increase the  **number of trials to 40  for the hyperparameter sweeps per model–attack pair** to reduce variance and stabilize the Pareto estimates. This produces a more realistic scenario in which the most harmful configuration is selected while respecting a bounded decrease in utility.

---

> ### Author Response · Authors · 2025-11-27
> **Reply to Reviewer Wc1L (Part 2)**
>
> ### (4) TamperBench evaluation metrics
>
> StrongREJECT and MMLU-Pro were selected because of their widespread use in recent open-weight reports (e.g., Llama [3], Qwen [4]) and their complementary strengths: MMLU-Pro targets _challenging reasoning tasks_, while StrongREJECT [9] provides a _robust refusal evaluator_, benchmarked against human annotators and specifically designed to detect harmful responses more reliably than traditional safety judges. These metrics allow us to measure tampering effects consistently across model families and attack types.
>
> Recognizing that safety and capability are multi-dimensional, we expand the evaluation in the revision. After running a suite **eight distinct tampering attacks** across both Qwen3-4B and Qwen3-4B-Base (which provide a range of capability drops), we run additional evaluations on the same checkpoints with:
> - **MATH** [5] (domain-specific reasoning with exact-match criteria),
> - **MBPP** (pass@1) [6], which evaluates _coding and program synthesis problem solving_,
> - **IFEval** [7], which targets _instruction-following compliance_, and
> - **JailbreakBench** [8], an independent safety jailbreak robustness benchmark.
>
> We then examine the relationships between these and our primary metrics, visualized in **Figure 7 (Appendix A.1)**. We observe that:
> - MMLU-Pro changes are **strongly correlated with IFEval and MBPP shifts**, suggesting that erosion of general reasoning and instruction-following often co-occurs with degradation in coding/problem-solving ability. A weaker correlation is observed with MATH, likely due to its narrow domain scope and exact-match criterion.
> - _StrongREJECT changes align closely with JailbreakBench_, indicating consistent detection of harmfulness across evaluators.
>
> These findings reinforce that, StrongREJECT and MMLU-Pro serve as **reasonable proxies for tampering-induced safety and capability shifts**, particularly given the substantial compute cost of running multiple high-fidelity benchmarks at scale. Importantly, **TamperBench is designed to be extensible**: additional metrics can be integrated via the evaluation registry with minimal effort (see **Appendix A.5.2**), enabling future work to incorporate domain-, modality-, or task-specific evaluators when appropriate.
>
> References:
> - [1] Model Tampering Attacks Enable More Rigorous Evaluations of LLM Capabilities (Che et al.) – https://arxiv.org/abs/2502.05209
> - [2] Fine-tuning Aligned Language Models Compromises Safety, Even When Users Do Not Intend To! https://arxiv.org/abs/2310.03693
> - [3] The Llama3 Herd of Models https://arxiv.org/abs/2407.21783
> - [4] Qwen Technical Report https://arxiv.org/abs/2309.16609
> - [5] Measuring Mathematical Problem Solving With the MATH Dataset https://arxiv.org/abs/2103.03874
> - [6] Instruction-Following Evaluation for Large Language Models https://arxiv.org/abs/2311.07911
> - [7] Program Synthesis with Large Language Models https://arxiv.org/abs/2108.07732
> - [8] JailbreakBench: An Open Robustness Benchmark for Jailbreaking Large Language Models https://arxiv.org/abs/2404.01318
> - [9]  A StrongREJECT for Empty Jailbreaks https://arxiv.org/abs/2402.10260

---

### Official Review · Reviewer_DvMH · 2025-11-01

**Soundness:** 3
**Presentation:** 2
**Contribution:** 2
**Rating:** 2
**Confidence:** 4

**Summary:**

This paper introduces TamperBench, a unified, extensible benchmark for evaluating the tamper resistance of open-weight large language models (LLMs). The authors include several existing benchmarks in helpfulness and safety evaluation such as MMLU-Pro, or Strongrejct and test 19 models in their unified benchmarks.

**Strengths:**

1. The evaluation of LLMs safety and helpfulness is crucial. And a unified framework to evaluate them could largely save the efforts in reproducing different env settings.

2. This paper include white-box, black-box,  latent-space representation, and fine-tuning attacks, which covers a wide range.

3. The presentation is easy to read and follow.

**Weaknesses:**

1. Lack of novelty. This paper seems to only combine existing helpfulness and safety evaluation metric together, and there are not new metrics or benchmarks.

2. I think the main contribution comes from the re-organization of existing benchmarks. However, some examples in this paper such as MMLU, StrongREJECT have well-structured open-source code, making them easily to employ and test. I don't think re-organize them have saved a lot of time costs rather than directly use their official code. This seems trivial.

3. Lack of human evaluation. For a safety evaluation benchmark, you should at least personally check the cases in case of false positive rate.

4. Apart from this re-organization, is there any other contributions or any other problems you find existing benchmarks have and solve in this paper?

**Questions:**

Please see the weakness.

---

> ### Author Response · Authors · 2025-11-27
> **Reply to Reviewer DvMH (Part 1)**
>
> We appreciate the reviewer’s feedback and the recognition of the breadth of attacks across different settings. We look to address their comments point by point.
>
> ---
>
> ### (1) Novelty
>
> We would like to clarify that the contribution does not lie in combining metrics. TamperBench introduces the **first standardized framework for evaluating weight-space and representation-space tampering**: implementing multiple tampering attacks, providing a common interface for defenses, and executing hyperparameter sweeps for attacks to produce reliable risk and safety evaluation. Metrics are the evaluators applied at the end of this process.  This a problem setting that existing benchmarks do **not** cover.
>
> Current benchmarks used in the community—e.g., **HarmBench** [1] and **PandaGuard** [2]—focus exclusively on _input-space jailbreaks_ (prompt-level attacks, etc.). They **do not evaluate model modification**, and therefore cannot assess safety erosion caused by fine-tuning (weight-space) [3, 4] or representation space modifications [5].
>
> To the best of our knowledge, there is **no existing benchmark** that:
>
> 1. Implements **a diverse suite of tampering attacks** spanning both weight-space and representation-space,
> 2. Combines these attacks with hyperparameter optimization to reduce variance and **reflect realistic adversarial search**
> 3. Benchmarks vulnerability to tampering across **21 open-weight models** highlighting insights.
>
> Recent works surveying the state of tampering and position papers on technical problems for open-weight LLMs [6, 7], explicitly call for such a benchmark, highlighting that the field lacks standardized benchmarking.
>
> ---
>
> ### (2) Contribution and Time Saving
>
> As mentioned before, to the best of our knowledge, _no existing benchmark compiles state-of-the-art weight-space and representation-space tampering attacks into a single, unified evaluation framework._ Moreover, several tampering attacks have **no open-source implementations** — including _few-shot harmful fine-tuning attacks_ (Che et al. [4]), _jailbreak-tuning / competing-objectives backdoor attacks_ (Murphy et al. [8]), and _multilingual fine-tuning attacks_ (Poppi et al. [9]). Others often exist only in _outdated codebases for embedding-space attacks_ (Schwinn et al. [10]) that are incompatible with modern `transformers` releases, preventing evaluation on current models such as **Qwen3**. TamperBench is therefore **the first end-to-end toolkit** to implement these attacks in a maintained, interoperable form factor.
>
> TamperBench fills this gap by providing:
> - Unified **attack and evaluation registry**,
> - **Optuna-based sweep functionality** (grid, single-objective, and bi-objective) to reflect realistic adversarial search,
> - **Checkpointing and logging** across trials, and
> - **Common evaluation stack** (StrongREJECT [11], MMLU-Pro [12], and—per the revision—additional benchmarks including IFEval [13], MATH [14], MBPP [15], and JailbreakBench [16]; **Appendix A.1**).
>
> The contribution is not metrics aggregation: StrongREJECT and MMLU-Pro are **evaluators applied at the end of a broader experimental pipeline**. The core novelty lies in making diverse _tampering attacks operational, comparable, and extensible_, and providing a reliable methodology for systematically stress-testing model tamper resistance. TamperBench significantly reduces engineering overhead and establishes a foundation for future tamper-resistance research.
>
> ---
>
> ### (3) Human evaluation
>
> In this work, we rely on StrongREJECT which has been benchmarked against human raters and prior judges in its original paper [11]. Our focus is on relative changes under tampering—how much measured harmfulness shifts for a previously validated judge.

---

> ### Author Response · Authors · 2025-11-28
> **Reply to Reviewer DvMH (Part 2)**
>
> ### (4) Other Contributions and findings
>
> We would like to clarify that TamperBench is not just a combination of metrics; as it **introduces provides the first standardized framework for weight-space and representation-space tampering evaluation**, comprising the implementation of attacks [4, 5, 6, 7, 8, 9, 10], in addition to the aggregation of metrics, and interfacing of tamper-resistant defenses like LAT [18] and TAR [17].
>
> TamperBench also enables **realistic adversarial settings**. Instead of using pre-defined hyperparameters (which are common in prior work [4, 7, 8]), every model–attack pair for fine-tuning attacks is evaluated using **Optuna-based sweeps**, to reliably stress-test tamper resistance.
>
> Finally, TamperBench enables comparison of models and defences, and by benchmarking 21 models across 9 attacks, we find consistent patterns across certain model-families as discussed in **Section 4.3** of the revised manuscript:
> - _Qwen3 base models are more tamper-able than their instruction-tuned variants_,
> - _Llama-3 instruction-tuned models become *more* vulnerable than their base counterparts_
> - _Qwen3-8B is marginally more tamper-resistant than other open-weight 7 - 8B models_ (not including defenses).
>
> References:
> - [1] HarmBench: A Standardized Evaluation Framework for Automated Red Teaming and Robust Refusal – https://arxiv.org/abs/2402.04249
> - [2] PANDAGUARD: Systematic Evaluation of LLM Safety in the Era of Jailbreaking Attacks
> - [3]  Fine-tuning Aligned Language Models Compromises Safety, Even When Users Do Not Intend To! https://arxiv.org/abs/2310.03693
> - [4] Model Tampering Attacks Enable More Rigorous Evaluations of LLM Capabilities (Che et al.) – https://arxiv.org/abs/2502.05209
> - [5] Revisiting the Robust Alignment of Circuit Breakers https://arxiv.org/abs/2407.15902
> - [6] Harmful Fine-tuning Attacks and Defenses for Large Language Models: A Survey: https://arxiv.org/abs/2409.18169
> - [7] Open Technical Problems in Open-Weight AI Model Risk Management https://papers.ssrn.com/sol3/papers.cfm?abstract_id=5705186
> - [8] Jailbreak-Tuning: Models Efficiently Learn Jailbreak Susceptibility (Murphy et al.) – https://aclanthology.org/2025.emnlp-main.669/
> - [9] Towards Understanding the Fragility of Multilingual LLMs against Fine-Tuning Attacks https://arxiv.org/abs/2410.18210
> - [10] Soft Prompt Threats: Attacking Safety Alignment and Unlearning in Open-Source LLMs through the Embedding Space  https://arxiv.org/abs/2402.09063
> - [11] A StrongREJECT for Empty Jailbreaks https://arxiv.org/abs/2402.10260
> - [12] MMLU-Pro: A More Robust and Challenging Multi-Task Language Understanding Benchmark https://arxiv.org/abs/2406.01574
> - [13] Instruction-Following Evaluation for Large Language Models https://arxiv.org/abs/2311.07911
> - [14] Measuring Mathematical Problem Solving With the MATH Dataset https://arxiv.org/abs/2103.03874
> - [15] Program Synthesis with Large Language Models https://arxiv.org/abs/2108.07732
> - [16] JailbreakBench: An Open Robustness Benchmark for Jailbreaking Large Language Models https://arxiv.org/abs/2404.01318
> - [17] Tamper-Resistant Safeguards for Open-Weight LLMs. https://arxiv.org/abs/2408.00761
> - [18] Latent Adversarial Training Improves Robustness to Persistent Harmful Behaviors in LLMs  https://openreview.net/forum?id=6LxMeRlkWl

---

### Official Review · Reviewer_e3qj · 2025-11-02

**Soundness:** 2
**Presentation:** 1
**Contribution:** 2
**Rating:** 2
**Confidence:** 4

**Summary:**

This paper identifies a key gap in large language model (LLM) safety research: the lack of standardized evaluation for tamper resistance. The authors argue that current methods for testing defenses against model modifications (like fine-tuning attacks) are ad-hoc, using varied datasets, metrics, and threat settings, which makes fair comparison impossible.

To solve this, they introduce TamperBench, a unified and extensible benchmark framework designed to systematically stress-test the safety of open-weight LLMs against tampering.

Overall, the idea behind this paper is good, and I think it could become a strong benchmark, but due to the number of weaknesses identified I cannot recommend it for acceptance at this time. If the authors address my concerns, I would be willing to raise my score.

**Strengths:**

- Even though there are some key missing citations (see below), this paper is fairly extensive with its citations.
- The paper develops a standardized evaluation framework for an important class of attacks (representation-space and weight-space attacks), which differ from the input-space attacks most often considered in LLM robustness evaluations.
- Hyperparameter sweeps for the fine-tuning attacks is a crucial consideration (although see concern below about needing different optimizers). Too many papers in this area only consider one or a small number of fine-tuning attacks.

**Weaknesses:**

Weaknesses:
- For the fine-tuning attack hyperparameters, it would have been good to consider different optimizers as well as different hyperparameters.
- I don't know if defending against weight-space and representation-space attacks should be grouped together as "tamper-resistance". Tamirisa et al. introduced the term tamper-resistance in the context of fine-tuning attacks, and it seems useful to maintain precision in terminology (even though some latent space attacks can be considered subsets of fine-tuning attacks).
- Figure 5 is indecipherable. StrongREJECT and MMLU-Pro are mixed together in the figure with no clear distinction, it's not clear which numbers are better, and the color coding on the right side of the plot have two scales each with no explanation!
- Why are actual methods that try to improve tamper-resistance and latent space robustness not benchmarked? E.g., there are no numbers for LAT or TAR.
- Section 2 is confusingly structured. It would be cleaner to structure it as different "threat models" that are considered in TamperBench, each one clearly stating the assumptions, knowledge/resources, and goals of the attacker and defender. Currently it uses nonstandard terms like "threat setting" and "tamper-resistance goals and defenses". Granted, non-adversarial/accidental tampering wouldn't fit into a threat model framing. Overall, though, section 2 could be presented more clearly.
    - For example, Section 2.3 is about the gap in the literature that TamperBench seeks to fill. This belongs in a related work section, not alongside the threat models! Readers won't expect to find it here, which makes following along harder.
- I don't really agree with the main taxonomy in Figure 3. Why is malicious tampering described as "disregards API protections". Isn't the most common setting here fine-tuning open-weight models? If so, it's kind of weird to describe this as disregarding API protections, since there is no API to speak of. I wouldn't want the community adopting this taxonomy as currently written. It might make more sense to have a 2x2 grid: intentional vs unintentional tampering and API vs open-weight tampering. It seems like those are the two relevant axes being described in this paper. Maybe a third axis is overt vs covert tampering.

Missing citations:
- "Self-Destructing Models: Increasing the Costs of Harmful Dual Uses of Foundation Models" from Peter Henderson et al. (a key early paper that I'm surprised is not cited)
- "Eternal Sunshine of the Spotless Net: Selective Forgetting in Deep Networks" and "Forgetting Outside the Box: Scrubbing Deep Networks of Information Accessible from Input-Output Observations" from Aditya Golatkar et al. (some of the first papers describing resistance to fine-tuning attacks as a key metric for unlearning)

Suggestions (not counting toward score):
- It would be interesting to discuss the continued relevance of API-based tampering attacks, since AI companies are really focusing on continual learning and long-term memory, which may end up being addressed with something like per-user PEFT.
- Please use `` and '' for quotations in LaTeX. E.g., "may emerge when ”safe”" on line 155 uses incorrect quotations.

**Questions:**

See weaknesses

---

> ### Author Response · Authors · 2025-11-27
> **Reply to Reviewer e3qj (Part 1)**
>
> We thank the reviewer for recognizing the importance of a standardized framework for tamper-resistance evaluation and for noting the need for considering attacks beyond input-space jailbreaks. We look to address each concern in detail below.
>
> ---
>
> ### (1) Optimizers in fine-tuning hyperparameter sweeps
>
> We agree that optimizer choice is a potentially impactful aspect of attack configuration, and as per the reviewer’s suggestion we evaluate it directly. We ran an extended hyperparameter sweep for the harmful LoRA attack on **Llama-3.1-8B-Instruct** for 70 trials, including **AdamW, SGD, and AdaFactor** in the search space, and we also repeated the sweep with AdamW only. As shown in **Section A.3** and **Figure 9** of the revised paper, introducing additional optimizers did not increase achievable attack strength: the strongest configurations sampled by Optuna were AdamW-based, and the **top five StrongREJECT scores were all obtained with AdamW**.
>
> While SGD/AdaFactor did not expand the harm envelope in this setting, we observed that the optimizer parameter remained highly ranked in Optuna’s importance estimates, indicating that it is a meaningful hyperparameter to explore. TamperBench offers optimizer choice as a configuration option, and adding further optimizers is straightforward through the attack registry should future work wish to investigate broader variants.
>
> ---
>
> ### (2) Grouping of weight-space and representation-space attacks
>
> We adopt the definition in Che et al. [1], where tampering is defined as _modifications to model parameters or latent representations_, which covers a broader class of manipulations and also encompasses the fine-tuning-based definition used in Tamirisa et al. [2] as a type of tampering within this definition. This allows us to treat weight-space and representation-space attacks in a unified way while acknowledging that some prior work focuses on a narrower subset. We **update the caption in Figure 1** to explicitly state the definition we use.
>
> ---
>
> ### (3) Figure 5 readability
>
> To improve clarity and readability of results, in the revision we introduce **Figure 6**, which shows **only StrongREJECT scores** _under constrained shifts in MMLU-Pro accuracy_. This isolates harmfulness changes and makes comparisons across models and defenses more transparent; **darker cells indicate increased harmfulness**, whereas lighter cells indicate lower harmfulness, and thus greater tamper-resistance (e.g., if a column corresponding to a model is lighter than another, it is likely more tamper-resistant, for instance: Qwen3-0.6B has lighter cells than Qwen3-0.6B-Base in **Figure 6**, and can therefore be interpreted as _relatively more tamper-resistant_ - a trend that holds for all Qwen base and post-trained models).
>
> We also updated **Figure 5** to improve its readability: it now focuses on **utility drop and harmfulness change alongside baselines**, uses more intuitive colour schemes (based on their presence in prior safety benchmark works [3]), and includes visual guides such as marked reference lines and additional labels. The revised version also illustrates a key pitfall of selecting **purely the highest-harm configuration**: some extreme points achieve large StrongREJECT increases at the cost of severe capability collapse, motivating the constrained selections shown in **Figure 6**.

---

> ### Author Response · Authors · 2025-11-27
> **Reply to Reviewer e3qj (Part 2)**
>
> ### (4) Missing LAT / TAR defenses
>
> In the revised version, we add **LAT** and **TAR** as defense-augmented variants of **Llama-3-8B-Instruct** and evaluate them under our main fine-tuning and representation-space attacks using the same sweep protocol as for other defenses. Results are reported alongside ReFAT, Circuit Breaking, and Triplet in **Figure 5** and **Figure 6**. At a high level, both LAT and TAR remain vulnerable to most attacks; however, TAR shows somewhat reduced harmfulness increases, albeit at the cost of lower utility.
>
> ---
>
> ### (5) Clarity of Section 2 and threat models / taxonomy
>
> We revise the writing and organization of these sections as follows: **Section 2 now serves only as background and related work**, while the **formal threat model has been moved to Section 3.1**, where attacker intentions, defender objectives, and tampering regimes are stated explicitly. The threat model now distinguishes regimes primarily along **(i) intent (benign vs. malicious)** and **(ii) access (API-based vs. open-weight)**, which aligns with the reviewer’s suggestion and avoids the terminology ambiguity noted in the original submission.
>
> **Figure 3 has been redesigned** to remove the previous “disregards API protections” phrasing. The revised figure presents regimes along **intent and overt/covert behaviour**, with API-based and open-weight settings annotated rather than conflated. This more clearly reflects a 2×2 framing and avoids forcing non-adversarial or accidental tampering into a purely adversarial template.
>
> ---
>
> ### (6) Missing citations (Henderson et al., Golatkar et al.)
>
> We have added citations to **Henderson et al., “Self-Destructing Models”** and **Golatkar et al.** on selective forgetting and unlearning, to acknowledge this early work on model resilience and fine-tuning resistance.
>
> References:
> - [1] Model Tampering Attacks Enable More Rigorous Evaluations of LLM Capabilities. https://arxiv.org/pdf/2502.05209
> - [2] Tamper-Resistant Safeguards for Open-Weight LLMs. https://arxiv.org/abs/2408.00761
> - [3] SORRY-Bench: Systematically Evaluating Large Language Model Safety Refusal. https://arxiv.org/abs/2406.14598
> - [4] Latent Adversarial Training Improves Robustness to Persistent Harmful Behaviors in LLMs  https://openreview.net/forum?id=6LxMeRlkWl

---

### Author Response · Authors · 2025-12-03

## Summary of Rebuttals and Revisions

---

We thank the reviewers and the AC for reviewing our paper and providing valuable feedback. We have aimed to tackle each individual concern, running additional experiments, expanding evaluation, and improving visual clarity and saliency of the work. Below we summarize our main revisions.

---

### **1. Expanded Safety & Capability Evaluation (4 New Evaluators Added)**
Reviewers raised concerns that StrongREJECT and MMLU-Pro may be narrow proxies for safety and capability. In the revision, we strengthened the evaluation by adding **four additional benchmarks**, each targeting a distinct capability axis or safety evaluation:
- **MATH** (domain-specific mathematical reasoning) [1],
- **MBPP** (coding & program synthesis; pass@1) [2],
- **IFEval** (instruction following) [3], and
- **JailbreakBench** (jailbreak robustness) [4].

We evaluated these on eight tampering attacks for Qwen3-4B / Qwen3-4B-Base and report correlations in **Figure 7**. We observe that StrongREJECT **aligns strongly** with JailbreakBench  and MMLU-Pro **correlates strongly** with MBPP and IFEval. These results show that _the original metrics are reasonable and representative proxies_, especially given the compute required for full multi-metric evaluation.

---

### **2. Selecting The Best Hyper-parameter Configuration**
To address concerns about selection of the maximum-StrongREJECT configuration, for every model–attack pair we now report both:
- The **unconstrained maximum-harm point**, and
- The **maximum-harm point, bounded by utility decrease** (≤10% MMLU-Pro drop), configurable and visualized in **Figure 6**.

We also increased each Optuna sweep to **40 trials**, stabilizing the Pareto front and better reflecting realistic attacker search.

---

### **3. Toolkit Contribution, Novelty, & Extensibility**
We more explicitly highlighted TamperBench's utility as a **technical toolkit** as well, highlighting the:
- _Implementation of state-of-the-art tampering attacks_—several of which **had no open-source implementation** before TamperBench.
- _Modernized and maintained implementations_ of attacks whose prior codebases are incompatible with current Transformers.
- _Third-party extensibility_, demonstrated in Appendix **A.5**, where researchers can subclass the attack interface, define search spaces, and automatically leverage sweeps, evaluations, and visualizations.

This clarifies that TamperBench provides **operational infrastructure**, enabling reproducible, comparable tampering research that previously required substantial engineering effort.

---

### **4. Paper Writing Improvements: Better Organization, Threat Model, and Taxonomy**
We updated the manuscript structure to improve clarity and alignment with reviewer expectations as per suggestions of Reviewer e3qj:
- **Section 2** now strictly covers background and related work.
- The **formal threat model** is now in **Section 3.1**, cleanly specifying attacker/defender assumptions and access regimes.
- **Figure 3** has been redesigned into a clearer **(intent × access)** schema, with overt/covert distinctions annotated.

---

### **5. Additional Experiments: Larger Models, LoRA Ablations, and Optimizers**
We expanded empirical coverage in several ways:
- _Larger models:_ We added **Qwen3-32B** results (Appendix **A.2**), showing that TamperBench successfully evaluates substantially larger open-weight models.
- _Dataset-size ablation:_ For harmful LoRA, we ran a 70-trial sweep with larger harmful datasets (Appendix A.3, Figure 9).
- _Optimizer exploration:_ We added sweeps including **AdamW, AdaFactor, and SGD**, finding (Appendix A.3) that the strongest attacks remain AdamW-based, consistent with reviewer expectations.

---

### **6. Novel Insights Enabled by TamperBench**
Using the unified evaluation pipeline, we surfaced several **new model-family patterns** that were not documented prior to this work:
- _Qwen3 base models are more tamperable than their post-trained variants_
- _Llama-3 instruction-tuned models become more vulnerable_ than Llama-3 base_
- _Mistral-7B-Instruct often exhibits the highest vulnerability_ in the 7–8B range
- Across similar parameter scales, _Qwen3-8B appears marginally more resistant_ than others.

These insights illustrate the value of TamperBench as a tool for comparing tamper resilience in LLMs


### **References**
[1] MATH – https://arxiv.org/abs/2103.03874
[2] MBPP – https://arxiv.org/abs/2108.07732
[3] IFEval – https://arxiv.org/abs/2311.07911
[4] JailbreakBench – https://arxiv.org/abs/2404.01318

---

### Author Response · Authors · 2025-12-03
**Summary after review process change**

In light of the changes to the review process after the OpenReview de-anonymization issue, we would like to briefly summarize the original reviewer comments and how they were addressed in the rebuttal and revision. Reviewers broadly recognized **the value and necessity of a standardized framework for tamper-resistance evaluation**, highlighting the breadth of attacks, clarity of the threat framing, and _the potential for TamperBench to become a foundational benchmark for the field_. The main concerns centered on the scope of evaluation metrics (reviewers Wc1L and 47D8) and the interpretation of Pareto selection for optimal parameters (47D8, Wc1L). We responded by expanding the evaluator suite beyond StrongREJECT and MMLU-Pro to **include four additional benchmarks** (MATH, MBPP, IFEval, JailbreakBench), and reporting both unconstrained and ≤10% utility-bounded tampering configurations with 40-trial sweeps to better reflect realistic adversarial search.

We also ran **additional experiments on larger models** (Qwen3-32B) and added **dataset-size and optimizer ablations** at reviewer request. Concerns regarding clarity of the paper—particularly around the threat model, taxonomy, and citation gaps (reviewer e3qj)—were addressed by restructuring Section 2 strictly as background/related work, moving the formal threat model to Section 3.1, and redesigning Figure 3 and Figure 5. We also incorporated canonical references. These structural revisions are visible in the updated manuscript, _marked in blue_, alongside expanded experimental material in **Appendix A.1** (additional evaluators), **A.2** (larger models), and **A.3** (ablations), with Pareto-bounded configurations summarized in **Figure 6**.

Given the initially positive elements noted by reviewers and the substantial revisions addressing each concern—through additional experiments, expanded evaluation, and clearer exposition—we hope the AC will weigh these changes accordingly, especially given that a reviewer (e3qj) explicitly mentioned that they would be willing to raise the score if concerns were addressed.

---

### Meta-Review · Area_Chair_j7FU · 2026-01-07

**Summary:**

While the reviewers acknowledge the significance of evaluating LLM tamper-resistance, and the authors made substantial revisions during the rebuttal phase (e.g., adding metrics, conducting experiments on 32B models, and refining the taxonomy), this work still does not meet the standards for acceptance. The primary reasons are as follows:
- Limited Novelty and Contribution: Multiple reviewers noted that this work is essentially an engineering consolidation of existing attacks and evaluation metrics. The paper does not propose new attack algorithms, novel defense theories, or original datasets. The empirical insights provided are largely model-specific and lack profound, generalizable principles.
- Methodological Limitations: Although the authors introduced Utility-Bounded evaluation, the framework relies solely on automated metrics from prior work (e.g., StrongREJECT) as proxies. The absence of human evaluation or necessary benchmark validation to calibrate false positive risks undermines the rigor of the methodology.
- Limited Scope of Attacks and Defenses: The analysis is confined to existing attack methods, and the inclusion of defenses appears to be a simple aggregation that remains incomplete. This contradicts the paper's claim of presenting a "unified and systematic" framework for tamper-resistance evaluation.
- Concerns regarding Generalizability and Scale: While experiments on 32B models were added, the study still excludes models at the 70B+ scale. This limits the verification and generalizability of the conclusions regarding large-scale models.

**Reviewer Concerns:**

Concerns Addressed in the Rebuttal
The authors conducted extensive additional experiments, resulting in significant improvements in the following areas:
- Resolution of confusion regarding concepts, taxonomy, and result presentation: The authors restructured the paper, clearly distinguishing between "weight space" and "representation space," and redefined the threat model, thereby eliminating the conceptual confusion present in the initial version. Furthermore, the reorganization of experimental results has improved clarity and intuitiveness.
- Partial revision of evaluation methods and metrics: The introduction of the Utility-Bounded Harmfulness Score addressed the issue of reporting only maximum harmfulness while ignoring catastrophic collapse in general model capabilities. Additionally, benchmarks such as MATH, MBPP, and IFEval were supplemented.
- Partial supplementation of experiments: The authors added comparative experiments using optimizers such as SGD and included evaluations of tamper-resistance defense methods (e.g., LAT, TAR).

Outstanding Concerns
The aforementioned improvements did not resolve the reviewers' core concerns, which remain significant and led to the decision to reject:
- Limited Novelty and Contribution: This is the primary reason for rejection. The work is essentially an engineering integration of existing attacks. The paper fails to propose new attack algorithms, novel defense theories, or original datasets, and does not provide profound insights that transcend specific model performance.
- Limitations in Experimental Scope: Even after the rebuttal additions, the models remain limited to small-to-medium parameter sizes (up to 32B) and still fail to cover ultra-large models (70B+). Furthermore, the analysis is confined to existing attack methods, and the inclusion of defenses appears to be a simple aggregation (which remains incomplete). This contradicts the paper's claim of presenting a "unified and systematic" tamper-resistance evaluation.
- Questionable Reliability of Evaluation: The concern raised by Reviewer DvMH regarding the risk of false positives inherent in relying solely on automated metrics (such as StrongREJECT) remains unresolved.

**Reviewer Scores:**

Reviewer e3qj's suggestions were specific, covering structural confusion, taxonomy errors, missing citations, and the need for optimizer comparisons. The authors addressed all these issues specifically in the rebuttal. Consequently, this reviewer would likely raise their score significantly.

Reviewer Wc1L's primary concerns included the singularity of metrics, the small size of fine-tuning datasets and models, and the issue of reporting only maximum harmfulness while ignoring the collapse of general model capabilities. The authors added 4 new metrics and experiments on 32B models, and introduced the Utility-Bounded Harmfulness Score, resolving the issue of unreasonable evaluation caused by utility collapse.

Reviewer 47D8 focused on the issues of "reporting only maximum harmfulness while ignoring utility collapse" and the "evaluation of stealthy attacks." Although the authors added the Utility-Bound metric, 47D8's core skepticism remains that this setup fails to capture the attacker's objective regarding stealthiness and does not align with realistic threat scenarios. Furthermore, the reviewer's concern regarding the limitations of attack methods was not resolved; the work remains confined to existing attack techniques, which contradicts the paper's claim of a general tamper-resistance evaluation.

Reviewer DvMH primarily questioned the paper's lack of novelty, the absence of human verification, and whether the improvements over existing work provide sufficient insights for future research. The authors were unable to address these fundamental issues in the rebuttal.

---

### Decision · Program_Chairs · 2026-01-26

Reject